 

# Structural basis for interdomain communication in SHIP2 providing high phosphatase activity

Johanne Le Coq[1], Marta Camacho-Artacho[1], José Vicente Velázquez[1], Clara M Santiveri[2], Luis Heredia Gallego[1], Ramón Campos-Olivas[2], Nicole Dölker[3], Daniel Lietha[1]*

[1]Cell Signalling and Adhesion Group, Spanish National Cancer Research Centre, Madrid, Spain; [2]Spectroscopy and Nuclear Magnetic Resonance Unit, Spanish National Cancer Research Centre, Madrid, Spain; [3]Structural Computational Biology Group, Structural Biology and Biocomputing Programme, Spanish National Cancer Research Centre, Madrid, Spain

**Abstract** SH2-containing-inositol-5-phosphatases (SHIPs) dephosphorylate the 5-phosphate of phosphatidylinositol-3,4,5-trisphosphate ($PI(3,4,5)P_3$) and play important roles in regulating the PI3K/Akt pathway in physiology and disease. Aiming to uncover interdomain regulatory mechanisms in SHIP2, we determined crystal structures containing the 5-phosphatase and a proximal region adopting a C2 fold. This reveals an extensive interface between the two domains, which results in significant structural changes in the phosphatase domain. Both the phosphatase and C2 domains bind phosphatidylserine lipids, which likely helps to position the active site towards its substrate. Although located distant to the active site, the C2 domain greatly enhances catalytic turnover. Employing molecular dynamics, mutagenesis and cell biology, we identify two distinct allosteric signaling pathways, emanating from hydrophobic or polar interdomain interactions, differentially affecting lipid chain or headgroup moieties of $PI(3,4,5)P_3$. Together, this study reveals details of multilayered C2-mediated effects important for SHIP2 activity and points towards interesting new possibilities for therapeutic interventions.

*For correspondence: dlietha@cnio.es

**Competing interests:** The authors declare that no competing interests exist.

## Introduction

The levels of soluble inositol phosphates and membrane integrated phosphoinositide lipids in the cell are regulated by inositol kinases and phosphatases. They play key roles in a variety of cellular processes, including cell proliferation, survival and vesicular trafficking. The generation of the phosphoinositide phosphatidylinositol 3,4,5-trisphosphate ($PI(3,4,5)P_3$) is triggered via stimulation of growth factor and cytokine receptors that activate type I phosphoinositide 3-kinase (PI3K) to produce $PI(3,4,5)P_3$ from $PI(4,5)P_2$. Upon stimulation, $PI(3,4,5)P_3$ levels peak rapidly followed by a prompt return to basal levels (*Stephens et al., 1993*; *Yip et al., 2008*). The two main enzymes responsible for $PI(3,4,5)P_3$ degradation are the tumor suppressor phosphatase and tensin homolog (PTEN), which removes the 3-phosphate (3 P) and the SH2-containing inositol 5-phosphatase (SHIP), which dephosphorylates the 5 P to generate $PI(3,4)P_2$. Deregulation of $PI(3,4,5)P_3$ levels is one of the most frequent causes of tumorigenesis (*Song et al., 2012*; *Vanhaesebroeck et al., 2012*).

The SHIP 5-phosphatase (5-Ptase) family consists of two members, SHIP1 and SHIP2. SHIP1 expression is limited to hematopoietic cells and spermatocytes, whereas SHIP2 is ubiquitously expressed (*Liu et al., 1998*; *Pesesse et al., 1997*). SHIP1 plays an important role in myeloid homeostasis and exhibits reduced levels or mutational inactivation in various leukemias and lymphomas,

suggesting a role as tumor suppressor by negatively regulating the PI3K/Akt pathway (*Brauer et al., 2012*; *Lo et al., 2009*; *Luo et al., 2003*). SHIP2 polymorphisms associate with susceptibility to type 2 diabetes mellitus and hypertension (*Hao et al., 2015*; *Kagawa et al., 2005*; *Kaisaki et al., 2004*) as well as opsismodysplasia (*Below et al., 2013*; *Huber et al., 2013*; *Iida et al., 2013*), a rare but severe type of skeletal dysplasia. SHIP2 knockout mice were reported to be resistant to dietary obesity (*Sleeman et al., 2005*) and mice expressing catalytically inactive SHIP2 displayed several developmental defects and lower insulin secretion (*Dubois et al., 2012*). Several studies associate SHIP2 with signaling triggered by growth factors, including EGF and FGF (*Jurynec and Grunwald, 2010*; *Olsen et al., 2006*). Like SHIP1, SHIP2 also plays important roles in cancer, although its behavior is more complex and tissue dependent. Similar to SHIP1, SHIP2 functions as a tumor suppressor in glioblastoma, erythroleukemia and squamous cell carcinoma (*Giuriato et al., 2002*; *Taylor et al., 2000*; *Yu et al., 2008*), but is oncogenic in breast carcinomas and colorectal cancer (*Hoekstra et al., 2016*; *Prasad et al., 2008*). On the one hand, this ambivalent oncogenic behavior of SHIP2 in different cell types likely depends on how efficient the SHIP2 product, PI(3,4)$P_2$, is further metabolized to PI(3)P, a reaction catalyzed by inositol polyphosphate 4-phophatases (INPP4). Accumulation of PI(3,4)$P_2$ in a case where the INPP4 reaction is rate limiting can be oncogenic, since PI(3,4)$P_2$ can still partially activate the Akt pathway (*Li and Marshall, 2015*), but is no longer accessible for degradation by PTEN. On the other hand, the SHIP product can be directly tumor suppressing by shutting down mTORC1 signaling (*Marat et al., 2017*).

SHIP1 and SHIP2 are large ~140 kDa multidomain proteins that share a very similar domain organization (*Figure 1A*). The SHIP Ptase domain belongs to a family of $Mg^{2+}$-dependent inositol phosphatases with specificity for 5 P dephosphorylation of inositol rings. These 5-Ptases have different substrate recognition specificities (*Jefferson and Majerus, 1996*) with the SHIP 5-Ptase exhibiting a strong preference for 3-phosphorylated substrates (*Chi et al., 2004*; *Pesesse et al., 1998*; *Trésaugues et al., 2014*). Further, the inositol 5-Ptase family displays distant homology to $Mg^{2+}$-dependent DNase I and Apurinic/Apyrimidinic (AP) endonucleases (*Dlakić, 2000*; *Whisstock et al., 2000*) and the conservation of catalytic residues suggests a common mechanism of catalysis (*Whisstock et al., 2002*). The existence of a C2 domain following the Ptase domain was previously predicted for SHIP1 (*Ong et al., 2007*), however, the sequence identity to other C2 domains is low and a C2 fold has not been structurally confirmed. The SHIP1 C2 domain was reported to allosterically upregulate Ptase activity by binding to the PI(3,4)$P_2$ product, a feature not found to be conserved in SHIP2.

Here, we report the crystal structure of SHIP2 containing the 5-Ptase and a C2 domain (Ptase-C2), revealing that the two domains tightly associate via an extensive interface. We show that, both the Ptase and Ptase-C2 regions of SHIP2 bind phosphatidylserine (PS), which likely localizes the rigid Ptase-C2 region close to its substrate. Importantly, even in the absence of a localization effect, the C2 domain provides activating signals to the Ptase active site, resulting in a significant increase in catalytic turnover. Combining enzyme kinetics, molecular dynamics (MD) simulations and mutagenesis we identify distinct regions at the domain interface with different effects on SHIP2 activity. Hydrophobic Ptase-C2 interactions increase SHIP2 turnover with the lipid chain containing PI(3,4,5)$P_3$ substrate, whereas polar interactions communicate from the domain interface to a substrate binding loop mainly affecting catalysis of the soluble inositol 1,3,4,5-tetrakisphosphate (IP$_4$) substrate. Together, our data provide novel structural and mechanistic insights on a C2 assisted functioning of SHIP2, and by homology likely of SHIP1.

## Results

### Structure of SHIP2 Ptase-C2

A construct of human SHIP2 containing residues 420–878 corresponding to the Ptase and a C2 domain (Ptase-C2) was crystallized and diffraction data collected to 1.96 Å. The protein crystallized in the P2$_1$2$_1$2$_1$ space group with eight molecules per asymmetric unit (for crystallographic and refinement statistics, see *Table 1*). Each of the eight molecules (A-H) contains a 5-Ptase and a C2 domain (*Figure 1B*). The overall structure of the Ptase domain in Ptase-C2 is similar to the one previously reported (*Mills et al., 2012*; *Trésaugues et al., 2014*), displaying a central β-sheet sandwich decorated by seven helices (α1-α7) and several loops (L1-L4) (*Figure 1B*). However, there are significant

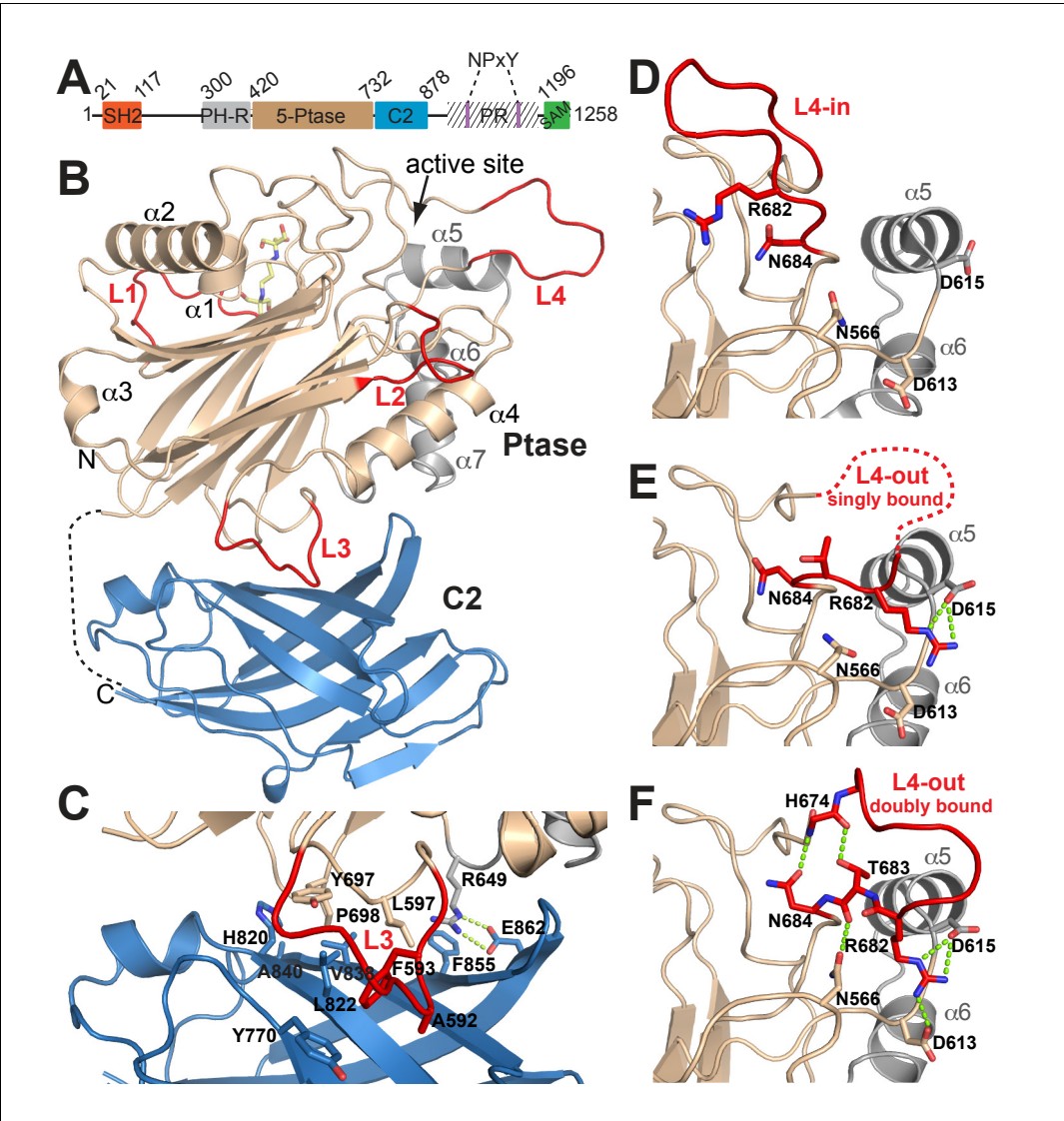

**Figure 1.** Structure of SHIP2 Ptase-C2. (A) Schematic domain structure of human SHIP2. SH2, Src homology domain 2; PH-R, pleckstrin homology related domain; 5-Ptase, 5-phosphatase; PR, proline rich; SAM, sterile-α-motif. (B) Ribbon representation of SHIP2 Ptase-C2, molecule B. The 5-Ptase domain is colored in tan and gray with loops (L1–L4) in red and the C2 domain in blue. The disordered linker is shown as dashed line and the site of catalysis is marked (active site). (C) Close-up of the domain interface. (D–F) L4 can switch between 'in' and 'out' conformations. The L4-in conformation, with R682 pointing towards the active site is seen in two Ptase crystal structures (shown in panel D is PDB 3NR8, chain B). The L4-out conformation is only seen in Ptase-C2 WT crystal structures, where R682 is either singly bound to D615 (panel E, shown is molecule G) or doubly bound to D613 and D615 (panel F, shown is molecule B).

The following figure supplements are available for figure 1:

**Figure supplement 1.** Details of interactions between the Ptase and C2 domains in Ptase-C2 WT.

**Figure supplement 2.** Details of interactions between the Ptase and C2 domains in Ptase-C2 FLDD.

**Table 1.** Diffraction and refinement statistics.

| | Ptase-C2 WT | Ptase-C2 FLDD | Ptase-C2 FLDD | Ptase-C2 D607A |
|---|---|---|---|---|
| Space group | $P2_12_12_1$ | $P2_1$ | $I2$ | $P2_12_12_1$ |
| Cell dimensions | | | | |
| a, b, c (Å) | 136.0, 175.8, 176.9 | 44.0, 81.1, 128.9 | 43.7, 73.4, 158.0 | 137.1, 177.1, 177.4 |
| α, β, γ (°) | 90.0, 90.0, 90.0 | 90.0, 92.9, 90.0 | 90.0, 90.7, 90 | 90.0, 90.0, 90.0 |
| Resolution (Å)* | 48.97–1.96 (1.99–1.96) | 81.12–1.94 (1.99–1.94) | 78.98–1.85 (1.89–1.85) | 49.17–2.65 (2.70–2.65) |
| $R_{merge}$* | 8.0 (88.1) | 13.3 (110.7) | 5.9 (59.5) | 16.1 (134.4) |
| $R_{meas}$* | 8.6 (95.6) | 14.4 (120.0) | 7.0 (70.4) | 17.0 (142.5) |
| $R_{pim}$* | 3.3 (36.8) | 5.5 (45.9) | 3.8 (37.2) | 5.5 (46.7) |
| CC (1/2)* | 0.999 (0.743) | 0.998 (0.787) | 0.999 (0.768) | 0.996 (0.700) |
| Mean (I/σ(I))* | 14.8 (2.1) | 11.8 (2.3) | 11.6 (2.1) | 13.4 (2.2) |
| Completeness (%)* | 100.0 (100.0) | 100.0 (100.0) | 99.3 (99.1) | 100.0 (100.0) |
| Multiplicity * | 6.8 (6.7) | 6.8 (6.7) | 3.4 (3.5) | 9.0 (8.9) |
| Refinement | | | | |
| Resolution (Å) | 48.96–1.96 | 81.12–1.94 | 78.98–1.85 | 49.17–2.65 |
| No. reflections | 287002 | 63790 | 40283 | 119204 |
| $R_{work}/R_{free}$ | 18.0/20.8 | 19.0/22.9 | 17.8/20.5 | 20.6/24.5 |
| No. atoms | | | | |
| Protein | 28142 | 6901 | 3423 | 27555 |
| Ligand | 225 | 39 | 16 | 237 |
| Water | 1564 | 288 | 139 | 284 |
| B- factors | | | | |
| Protein | 46.77 | 27.37 | 34.01 | 61.54 |
| Ligand/ion | 42.66 | 34.64 | 42.10 | 66.88 |
| Water | 44.01 | 29.82 | 38.30 | 45.57 |
| R.m.s. deviation | | | | |
| Bond lengths (Å) | 0.009 | 0.010 | 0.008 | 0.009 |
| Bond angles (°) | 1.306 | 1.354 | 1.260 | 1.267 |

*Highest resolution range shown in parentheses

differences. Loop residues 587–594 (L3) are located at the interface with the C2 domain and are due to extensive interactions with the C2 domain well ordered (*Figure 1C*). Importantly, a loop proximal to the active site, spanning residues 674–684 (L4), was in previous Ptase structures disordered or closed over the active site (*Mills et al., 2012*; *Trésaugues et al., 2014*) (*Figure 1D*; hereafter referred to as L4-in conformation), whereas in our structure 6 of the Ptase-C2 monomers (A, B, C, E, F, G) exhibit a conformation where L4 points away from the active site (L4-out). These L4-out conformations are stabilized by interactions of R682 in L4 with one or two aspartic acids (D613/D615) adjacent to helix α5 (*Figure 1E–F*, hereafter referred to as singly or doubly bound L4-out conformations). In molecule D, R682 is half way between the L4-in and L4-out conformation and in monomer H R682 is disordered. Together, these conformations suggest extensive mobility of L4 with a displacement of the guanidinium group of the R682 side chain by ~20 Å between L4-in and L4-out conformations. L4-out conformations appear to be favored by the presence of the C2 domain, since they are only observed for Ptase-C2, although crystal packing analysis suggests ample space in Ptase crystals in this region. In L4-out structures where R682 is doubly bound to D613 and

D615 (molecules B and F), a network of hydrogen bonds is established that includes H674, T683 and N684 on L4, rigidifying the loop (*Figure 1F*).

The C2 domain interacts with the Ptase on a face that is opposite to the active site (*Figure 1B*). The Ptase and C2 domains share an extensive interface of 940 Å$^2$ (*Figure 1C*), resulting in stabilization of the Ptase (*Supplementary file 1*). The interface is constituted largely of hydrophobic interactions, with A592, F593, L597, Y697 and P698 in the Ptase and Y770, H820, L822, E836, V838 and A840 in the C2 domain forming a hydrophobic core (*Figure 1C*, *Figure 1—figure supplement 1*). A significant peripheral polar contact is formed by R649 in the Ptase, forming hydrogen bonds with E862 and a cation-π interaction with F855 in the C2 domain. Notably, R649 is located in a loop that connects to a stretch of three helices (α5–7, colored grey in *Figure 1*), which at the other end join to D613 and D615, the docking site for R682 in the L4-out conformation.

## The C2 domain of SHIP2

Our structure confirms the presence of a β-sheet sandwich structure typical of C2 domains within residues 746–874 of SHIP2. Using the DaliLite server (*Holm and Rosenström, 2010*), we identified the closest structural homologues as the C2 domains of the itchy E3 ubiquitin protein ligase (pdb code 2NQ3, Z score = 12.7, RMSD over 106 residues = 2.1 Å) and dysferlin (pdb code 4IQH, Z score = 12.4, RMSD over 110 residues = 2.4 Å), which both belong to the PKC type C2 subfamily (*Zhang and Aravind, 2010*). A sequence-based similarity search using HH-pred (*Söding et al., 2005*) also identified the C2 domain of dysferlin as the closest related C2 domain (E-value = 0.049), although with 16% sequence identity the homology is low. The topology adopted by the SHIP2 C2 domain is as for itchy and dysferlin that of type II (or P-variant) (*Corbalan-Garcia and Gómez-Fernández, 2014*). PKC type C2 domains usually bind phospholipids, most frequently phosphatidylserine (PS), often in a Ca$^{2+}$-dependent manner via acidic residues present in three Ca$^{2+}$-binding loops (CBLs) at the tip of the C2 domain (*Figure 2A–B*) (*Cho and Stahelin, 2006*). Structure-based sequence alignment indicates that only 2 of 5 acidic residues on CBL1 and CBL3 are conserved between PKCα and SHIP (D829 and E832) and two are replaced by serines (S761 and S827). To test for lipid binding and calcium dependency, we performed surface plasmon resonance (SPR) experiments with immobilized PS vesicles and protein lipid overlay (PLO) assays. We found that both the Ptase and Ptase-C2 bind PS, with the C2 domain enhancing the interaction (*Figure 2C*, *Figure 2—figure supplement 1*). SPR experiments reveal that PS binding of Ptase-C2 exhibits a small increase of ~10% in presence of Ca$^{2+}$ (*Figure 2D*), an effect that was not detected in PLO experiments (*Figure 2—figure supplement 1*). In summary, we conclude that both, the Ptase and Ptase-C2 bind PS and the small effect of Ca$^{2+}$ suggests that the interaction is dominated by direct PS binding to basic residues, rather than Ca$^{2+}$-mediated binding to acidic residues and serines. Nevertheless, the low conservation of Ca$^{2+}$-binding residues in CBL1 and CBL3 of the SHIP2 C2 domain might weakly contribute to Ca$^{2+}$-mediated lipid binding (*Figure 2D*).

## The C2 domain affects SHIP2 activity

Next, we tested the effect of the C2 domain on SHIP2 activity. For this, we employed a Malachite Green assay to compare activity of SHIP2 Ptase-C2 to that of the isolated Ptase. Activity was measured with two soluble substrates, IP$_4$ and PI(3,4,5)P$_3$-diC8 and in all cases we observe a behavior close to conventional Michaelis-Menten enzyme kinetics (*Figure 3*). We find that the C2 domain increases turnover rates for both substrates (*Figure 3A–B*, *Table 2*). Interestingly, the effect is significantly larger with PI(3,4,5)P$_3$-diC8, where the C2 domain increases the k$_{cat}$ ~10 fold, while for IP$_4$ the increase is ~1.5 fold. As a result, the C2 domain switches the substrate preference with the Ptase alone turning over IP$_4$ twice as fast as PI(3,4,5)P$_3$-diC8, whereas Ptase-C2 displays ~ 4 fold faster kinetics with PI(3,4,5)P$_3$-diC8 compared to IP$_4$ (*Table 2*). The K$_M$ is less affected by the C2 domain with a significant change only with PI(3,4,5)P$_3$-diC8, where the absence of the C2 domain decreases the K$_M$ ~ 2 fold (*Table 2*). The fact that the distant C2 domain has specific and differential effects on catalysis of the two substrates, suggests the presence of an allosteric communication between the domain interface and the active site (*Figure 1B*).

The effect of the C2 domain was further investigated by mutagenesis. In order to disrupt the hydrophobic core of the domain interface we mutated F593 and L597 to aspartates (FLDD mutant). Further, we mutated R649 in the Ptase to alanine (R649A) to prevent polar interactions with E862

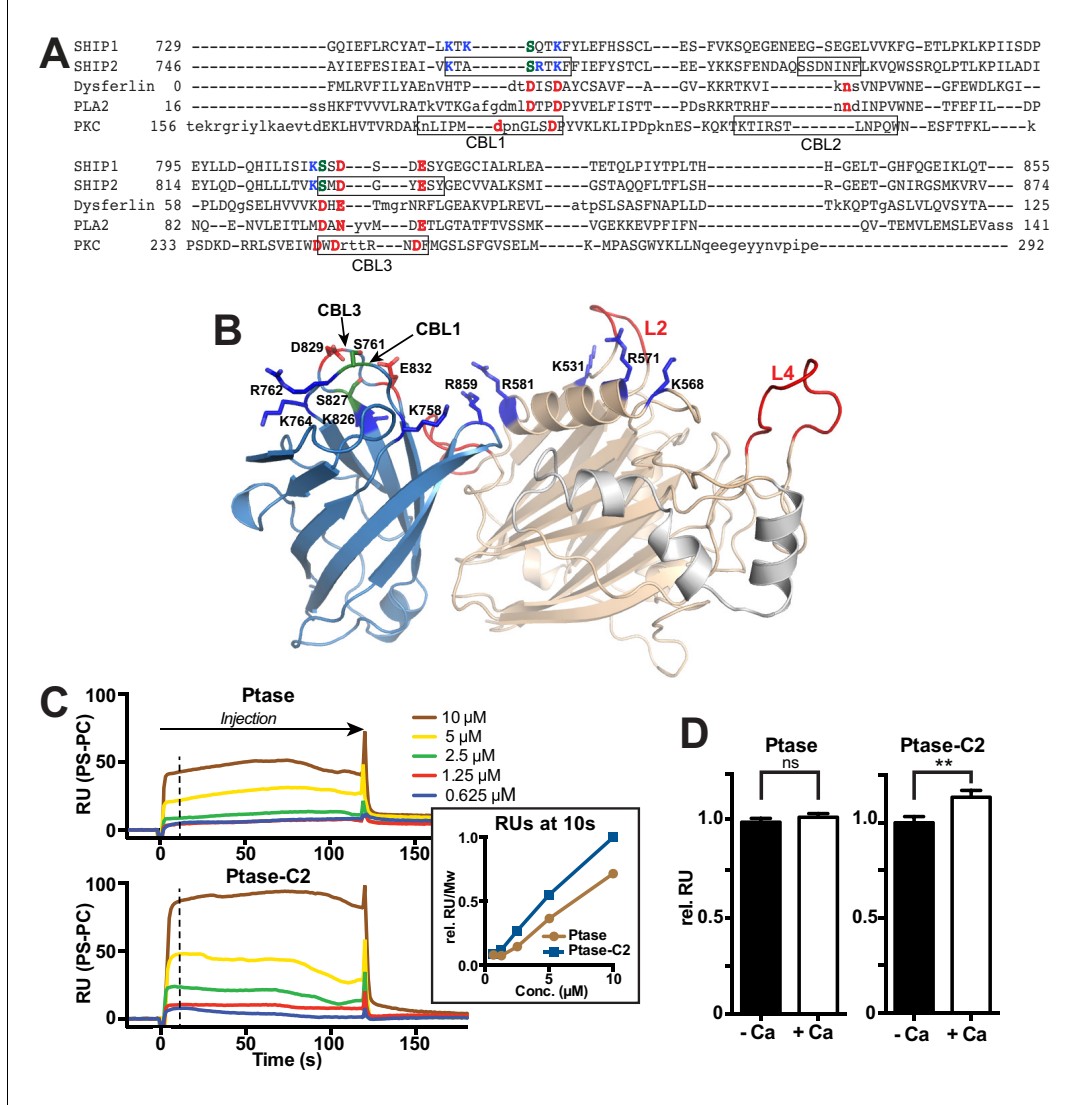

**Figure 2.** The C2 domain of SHIP2 and lipid binding. (**A**) Structure-based sequence alignment of the C2 domains of SHIP2, dysferlin, phospholipase A2 (PLA2) and protein kinase Cα (PKCα) and alignment of the corresponding sequence of SHIP1. Structurally equivalent positions with the SHIP2 C2 domain are in uppercase, insertions relative to SHIP2 are in lowercase. Calcium-binding loops (CBL) 1–3 of PKCα and corresponding loops in SHIP2 are boxed. Conserved acidic $Ca^{2+}$-binding residues and corresponding residues in SHIP are colored red, changes to serines are green and basic residues within or near the CBL's in SHIP are colored blue. (**B**) Putative lipid interactions in the SHIP2 Ptase-C2 region. Conserved acidic residues on CBL3 of SHIP2 (D829 and E832) are colored red and changes to serines green (S761 on CBL1 and S827 on CBL3). Basic residues on the Ptase and C2 domains expected to face the membrane are colored blue. For the mentioned residues, side chains are shown in stick representation. (**C**) SHIP2 binding to phosphatidylserine (PS) by surface plasmon resonance (SPR). SHIP2 Ptase and Ptase-C2 interactions to PS were studied using vesicles immobilized on a L1 sensor chip. Displayed are sensorgrams showing the difference in response between the active flow cell coated with 30% (mol/mol) PS vesicles and the reference cell containing phosphatidylcholine (PC) vesicles. The horizontal arrow indicates the association phase of Ptase and Ptase-C2 proteins, and the time axis is set to zero at the beginning of the injection. Insert: The SPR response units (RU) at 10 s of injection (dashed line), where we consider the steady state phase to be reached, is plotted. Values are corrected for the molecular weight (Mw) and relative to the highest response for Ptase-C2 at 10 μM. (**D**) Calcium dependency of the SHIP2 PS interaction. SPR responses of Ptase and Ptase-C2 were recorded as in panel C, but in presence or absence of 0.5 mM $CaCl_2$. Plotted are mean RUs relative to RUs in absence of $Ca^{2+}$ from triplicate injections of 5 μM protein and error bars represent SEM. ns: $p > 0.05$; **$p < 0.01$ (unpaired Student t test). See also.

The following source data and figure supplements are available for figure 2:

**Source data 1.** Source data for plots in *Figure 2C*-insert and 2D.

**Figure supplement 1.** Lipid binding by protein lipid overlay.

*Figure 2 continued on next page*

*Figure 2 continued*

**Figure supplement 1—source data 1.** Source data for plot in *Figure 2—figure supplement 1*.

and F855 in the C2 domain (*Figure 1C*). Crystal structures of Ptase-C2 FLDD confirm that few hydrophobic interactions remain intact between the Ptase and C2 domains (*Figure 1—figure supplement 2*). The FLDD mutations also disrupt interdomain hydrogen bonds between R649 and E862. When comparing the catalytic turnover of the FLDD mutants we find that the C2 domain no longer has a

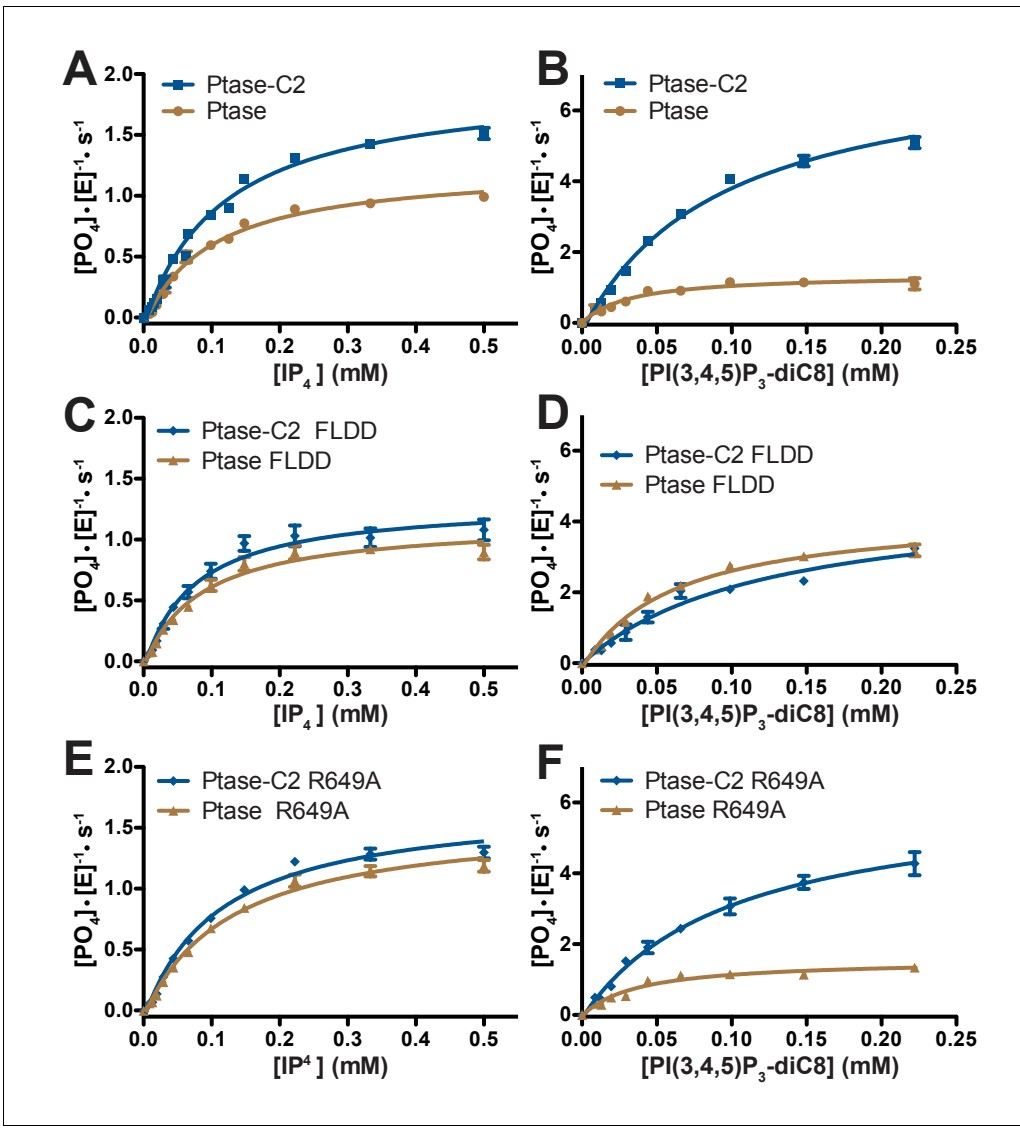

**Figure 3.** Enzyme kinetics of SHIP2. (**A–F**) Enzyme activity for SHIP2 Ptase and Ptase-C2 was measured using a Malachite Green assay, with $IP_4$ or $PI(3,4,5)P_3$-diC8 as substrates. Substrate titrations of wild type SHIP2 (**A–B**) and the interface mutants FLDD (**C–D**) and R649A (**E–F**) are shown. The enzyme concentration used in the shown plots is 400 nM (for $IP_4$) or 50 nM (for $PI(3,4,5)P_3$-diC8). Curves are fitted using the Michelis-Menten equation and derived $k_{cat}$ and $K_M$ values are shown in *Table 2*. Error bars represent SEM from at least three measurements.
The following source data is available for figure 3:

**Source data 1.** Source data for plots in *Figure 3A–F*.

**Table 2.** Enzymatic parameters are calculated by fitting the Michaelis-Menten equation to substrate titrations. Enzyme concentrations used for activity measurements were 400 nM, or if this caused saturated signals 50 nM (*).

| | Ptase | | | | Ptase-C2 | | | |
|---|---|---|---|---|---|---|---|---|
| | IP$_4$ | | PI(3,4,5)P$_3$ | | IP$_4$ | | PI(3,4,5)P$_3$ | |
| | $k_{cat}$ (s$^{-1}$) | $K_M$ (µM) | $k_{cat}$ (s$^{-1}$) | $K_M$ (µM) | $k_{cat}$ (s$^{-1}$) | $K_M$ (µM) | $k_{cat}$ (s$^{-1}$) | $K_M$ (µM) |
| WT | 1.32 ± 0.02 | 98 ± 7 | 0.69 ± 0.02 | 43 ± 4 | 2.02 ± 0.04 | 115 ± 8 | 7.83 ± 0.26* | 94 ± 9* |
| FLDD | 1.20 ± 0.05 | 82 ± 13 | 4.44 ± 0.17* | 61 ± 8* | 1.39 ± 0.06 | 73 ± 13 | 4.79 ± 0.49* | 122 ± 31* |
| R649A | 1.64 ± 0.05 | 126 ± 13 | 1.35 ± 0.06 | 59 ± 8 | 1.79 ± 0.05 | 110 ± 10 | 6.37 ± 0.42* | 100 ± 18* |

Source data 1. Source data for values shown in *Table 2*. All values are [PO$_4$] (in µM). Numbers (#*i*) above data indicate independent experiment number. Most kinetic parameters ($k_{cat}$, $K_M$) are extracted from curves shown in *Figure 3*, for which Source data are available with this figure. Below Source data are shown for cases where experiments were repeated at higher enzyme concentration (400 nM) to extract reliable kinetic parameters. Equation used to extract Vmax and $K_M$: Y = Bo + Vm*X/(X + $K_M$); Variables: Vmax, $K_M$, Bo = baseline. Software used: Graphpad Prism.

significant effect with either substrate (*Figure 3C–D*, *Table 2*), indicating that the FLDD mutations largely mimic the absence of the C2 domain. We note that the FLDD mutations increase intrinsic Ptase activity with PI(3,4,5)P$_3$-diC8, but importantly the presence of the C2 does not cause any further change. Interestingly, the polar R649A mutation specifically eliminates the effect of the C2 domain on IP$_4$ kinetics, while retaining most of its effect on PI(3,4,5)P$_3$ (*Figure 3E–F*). Together, these results suggest that the polar R649-E682/F855 interactions between Ptase and C2 domains mainly affect SHIP2 activity with IP$_4$, whereas hydrophobic interactions increase activity towards PI(3,4,5)P$_3$.

Next, we tested whether PS binding alters the C2-mediated effects. We find that soluble PS-diC8 has little effect on IP$_4$ kinetics and increases activity ~40% with PI(3,4,5)P$_3$-diC8 (*Figure 4A–B*), both for Ptase and Ptase-C2. If instead a lipid vesicle embedded PI(3,4,5)P$_3$-diC16 substrate is used, the presence of 30% (mol/mol) PS-diC16 in the vesicles increases SHIP2 activity ~2–3 fold for both, Ptase and Ptase-C2 (*Figure 4C*). The PS-mediated increase in activity on vesicles is maintained for the FLDD mutant, whereas as with soluble substrates, the increase in activity due to the C2 domain is abrogated by the interface mutations (*Figure 4D*). In summary, these data show that the effect of PS is amplified for SHIP activity on vesicles, possibly due to a localization or positioning effect and this effect is independent and additive to the enhancing allosteric signaling of the C2 domain.

## Communication between the C2 domain and the active site

In order to obtain insight on the path of communication between the C2 domain and the active site we performed unbiased molecular dynamics (MD) simulations of the SHIP2 Ptase in presence or absence of the C2 domain. Simulations were started in the doubly bound L4-out conformation, where R682 interacts with D613 and D615 (based on molecule B in the crystal structure). The simulations indicate that the C2 domain induces several dynamic changes in the Ptase domain, as can be seen from root mean square fluctuations (RMSF) throughout the simulation (*Figure 5A*). Not surprisingly, L3 is strongly stabilized due to its direct interaction with the C2 domain. On the other hand, L2 (residues 531–539) and L4 fluctuate stronger in presence of the C2 domain. RMSF analysis separated into different L4 states (R682 unbound, singly or doubly bound) indicate that higher L4 fluctuations occur when L4 is detached from D613/D615 (*Figure 5—figure supplement 1A*). Non-loop regions fluctuate little overall but fluctuations are increased in a stretch containing helices α5 and α6 and the proximal D613/D615 docking site of R682 (shaded grey in *Figure 5A–B*). Principal component analysis shows as one of the principal motions α5 moving away from α6 (*Figure 5A* insert, *Videos 1–2*). In order to analyze which regions deviate during the simulations most from the doubly bound L4-out starting conformation, we calculated root mean square deviations (RMSD) with respect to the L4-out starting structure, analyzed per residue and averaged over time. We find a significant difference in the region of α5, α6 and the proximal D613/D615, which in the presence of the C2 domain remain closer to the R682 doubly bound starting structure, whereas in absence of the C2 domain this region deviates significantly from the starting conformation (*Figure 5B*, also observed in

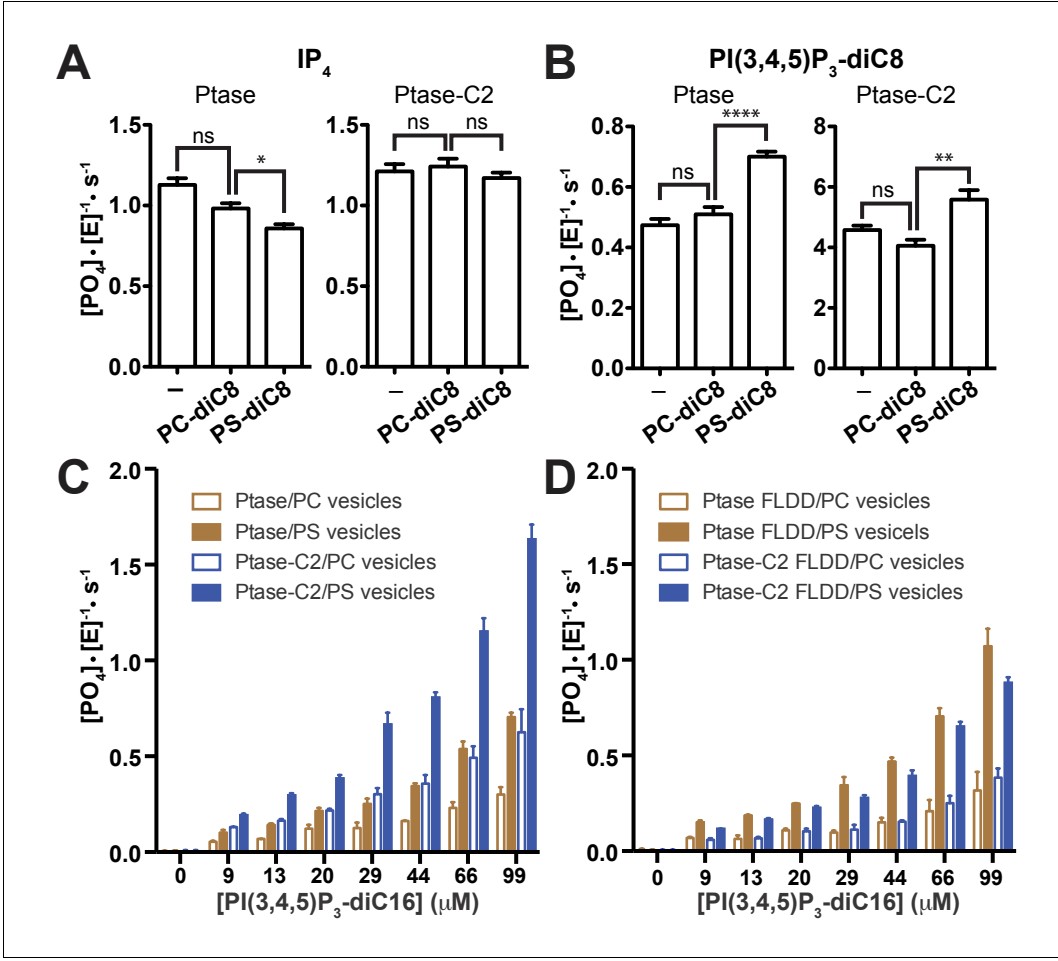

**Figure 4.** Effect of PS binding on SHIP activity. (**A–B**) Activity was measured in presence or absence of 100 µM PC-diC8 or PS-diC8 and as substrate 150 µM IP$_4$ (**A**) or PI(3,4,5)P$_3$-diC8 (**B**). Enzyme concentrations used were 400 nM for measurements with IP$_4$ (**A**) and 400 nM or 50 nM for Ptase and Ptase-C2 reactions, respectively, for measurements with PI(3,4,5)P$_3$-diC8 (**B**). ns: p>0.05; *p<0.05; **p<0.01; ****p<0.0001 (unpaired Student t test). (**C–D**) Activity was measured with the substrate PI(3,4,5)P$_3$-diC16 embedded in vesicles (10% mol/mol) that additionally contained 30% (mol/mol) PS or only PC. Enzyme concentrations used were 400 nM Ptase or 250 nM Ptase-C2 (**C**) or 400 nM of the FLDD interface mutants (**D**).

The following source data is available for figure 4:

**Source data 1.** Source data for graphs in *Figure 4A–F*.

*Videos 1–2*). The same RMSD analysis of different L4 states indicates that this occurs specifically in conformations where R682 is singly bound to either D613 or D615 (*Figure 5—figure supplement 1E*). Interestingly, different α5 helix positions are also observed in the Ptase-C2 crystal structure, confirming its mobility, and in agreement with MD simulations helix α5 is shifted specifically in singly bound R682 conformations by ~1.5 Å (*Figure 5C*). The C2 induced increase of L4 fluctuations when L4 is unbound (*Figure 5—figure supplement 1A*) together with stabilized D613/D615 when R682 is singly docked (*Figure 5—figure supplement 1E*) could potentially favor the transition from L4-in to L4-out and might be a reason why the L4-out conformation is only observed in Ptase-C2 crystal structures, but not in Ptase-only or Ptase-C2 FLDD structures.

On the other hand, once R682 detaches from D613/D615 we observe a more sustained release of L4 in MD simulations where the C2 domain is present. While in absence of the C2 domain R682 experiences only brief releases followed by immediate rebinding, in the Ptase-C2 simulation R682 remains fully detached from D613/D615 for ~1 µs of the 4.35 µs simulation (*Figure 5D*). Indeed,

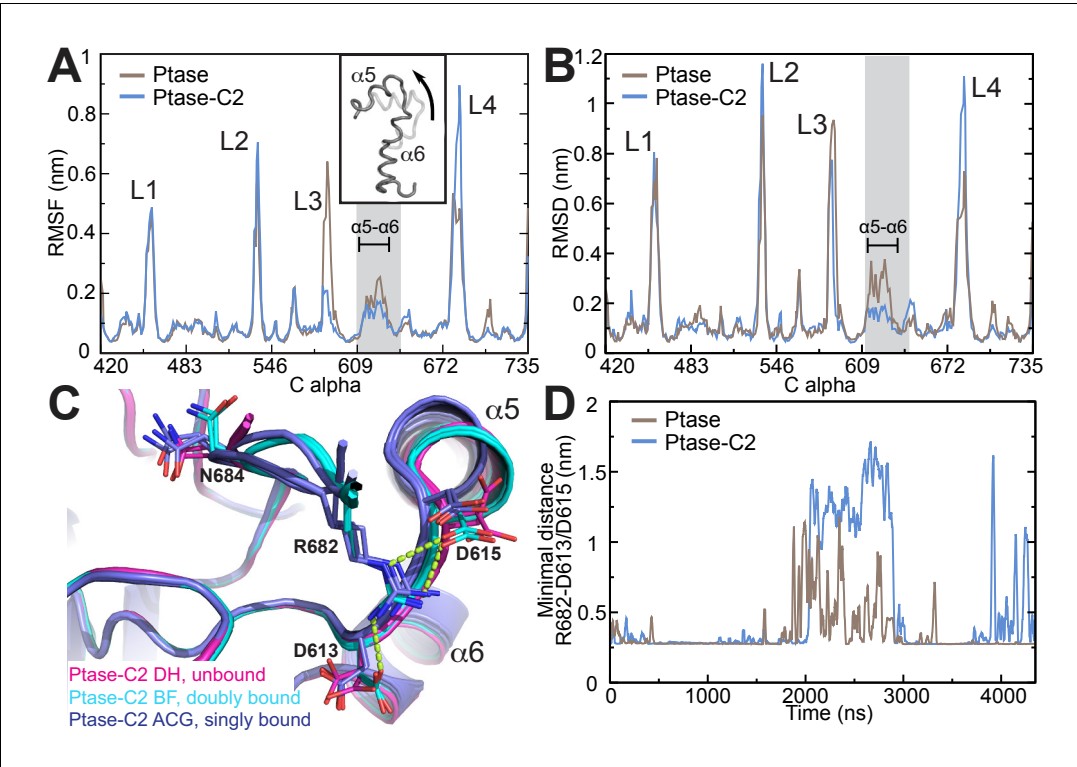

**Figure 5.** SHIP2 dynamics. (**A**) Root mean square fluctuations (RMSF) during MD simulations of the Ptase (brown) or Ptase-C2 (blue) are plotted for Cα atoms in the Ptase domain. Peaks correspond to loop regions (**L1–L4**) and the region corresponding to helices α5–7 is shaded gray with α5 and α6 indicated. Insert: Shown is the movement of helices α5 and α6 according to the first eigenvector from PCA analysis. (**B**) Root mean square deviations (RMSD) with respect to the L4-out starting structure, analyzed per residue and averaged over time is shown for Cα atoms in the Ptase domain. (**C**) Superposition of SHIP2 Ptase-C2 WT structures. R682 unbound structures are shown in magenta (molecules D and H), R682 doubly bound molecules in cyan (molecules B, F) and R682 singly bound molecules in blue (molecules A, C, G). Structures with singly bound R682 display displacement of helix α5. Doubly bound structures exhibit a N684 'up' conformation. Hydrogen bonds between R682 and D613/D615 are shown as light green dashed lines for molecule B. Molecule E is not included since it exhibits an alternate D613 conformation for the singly and doubly bound conformations with stronger occupancy for the singly bound state. (**D**) Plotted are the minimal distances between R682 and D613/D615 throughout the Ptase (brown) and Ptase-C2 (blue) simulations.

The following figure supplement is available for figure 5:

**Figure supplement 1.** SHIP dynamics for different L4 states.

---

cluster analysis of snapshots from the simulation reveals that for the Ptase alone all of the main clusters remain in a L4-out conformation, whereas for Ptase-C2 the third most populated cluster has R682 detached from D613/D615 (*Figure 5—figure supplement 1G–H*). Together, these data support a communication between the C2 domain and the active site in the Ptase by modulating the dynamics of helices α5–7 and L4 to facilitate transitions between L4-in and L4-out conformations. Structurally, this suggests a communication via R649, since this residue connects helices α5–7 to the C2 domain.

To understand how L4 dynamics could affect catalysis, we next analyzed the mode of substrate binding in SHIP2. Co-crystallization experiments using the catalytically inactive SHIP Ptase-C2 D607A mutant with $Mg^{2+}$ and substrate failed to reveal clear electron density for the substrate. Therefore we generated a model based on the homologue INPP5B crystal structure containing the PI(4)P product (*Trésaugues et al., 2014*) and insights from AP endonucleases, which are thought to share a common phosphate hydrolysis mechanism (*Whisstock et al., 2002*). We modeled the substrate

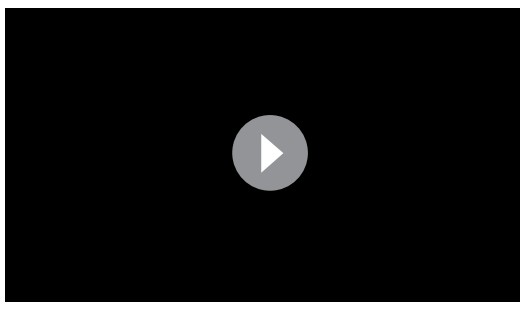

**Video 1.** Principal component analysis of Ptase simulation. Movements according to the first four eigenvectors from the principle component analysis of the Ptase simulation are shown. The starting conformation in the doubly bound L4-out conformation, based on molecule B of the Ptase-C2 crystal structure, is superimposed and colored blue.

according to a Michaelis-Menten complex, where all catalytic residues are in place for catalysis (*Figure 6A–B*). In this complex the catalytic base D607 stabilizes a water molecule for 5 P attack. According to our model a key 4 P interacting residue is N684 on L4. N684 can adopt two conformations (*Figures 1D–F* and *5C*) and needs to point towards the active site to interact with the substrate 4 P, which is seen in unbound or singly bound, but not doubly bound L4 conformations. In our model L4 adopts the L4-in conformation seen in 2 SHIP2 Ptase structures (PDB: 3NR8 and 4A9C; *Mills et al., 2012*; *Trésaugues et al., 2014*). In this conformation R682 is ideally positioned to interact with the 3 P of PI(3,4,5)P$_3$. We performed unbiased MD simulations of SHIP Ptase or Ptase-C2 starting with the PI(3,4,5)P$_3$-diC8 or IP$_4$ substrates bound according to our model shown in *Figure 6A–B*. Interestingly, the C2 domain significantly restricts the displacement of the headgroup of both substrates away from the Michaelis-Menten complex (*Figure 6C–D*, *Videos 3–4*).

Although our co-crystallization experiments did not reveal bound substrate they did exhibit electron density corresponding to a Mg$^{2+}$ ion and a phosphate group (*Figure 6—figure supplement 1A*). The Mg$^{2+}$ ion is on average displaced by 1.3 Å and the phosphate by 1.7 Å compared to the Mg$^{2+}$ and 5 P positions in our model (*Figure 6—figure supplement 1B*). The position of the phosphate does not leave space for an attacking water bound to D607, hence this could represent the post-catalysis positions of the Mg$^{2+}$ and 5 P.

## Probing the communication path by mutagenesis

The simulations and crystal structures indicate that a potential allosteric path linking the C2 domain to the active site could lead via R649 to helices α5–7 and at the other end of the helices through D613 and D615 to R682 and L4. To test this hypothesis, we generated the following mutants along the path: R649A (already generated as an interface mutant), R691A, D613A/D615A, R682A, N684A as well as active site mutants D607A and R665A (*Figure 7A*). All mutants were generated for Ptase and Ptase-C2 with the exception of R691A and D613A/D615A, which appear to destabilize the iso-

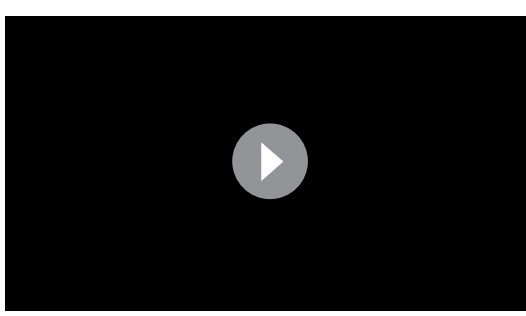

**Video 2.** Principal component analysis of Ptase-C2 simulation. Movements according to the first four eigenvectors from the principle component analysis of the Ptase-C2 simulation are shown. The starting conformation in the doubly bound L4-out conformation, based on molecule B of the Ptase-C2 crystal structure, is superimposed and colored blue.

lated Ptase and only yielded soluble protein for Ptase-C2. Circular dichroism (CD) analysis indicates that most of the mutations do not alter protein structure, apart from R691A in Ptase-C2 and N684A in the Ptase (*Figure 7—figure supplement 1*). The mutants were tested for changes in activity with IP$_4$ and PI(3,4,5)P$_3$-diC8 (*Figure 7*, *Table 3*). Interestingly, D613A/D615A mutated Ptase-C2 behaves similar to the R649A mutant, in that both only affect turnover with IP$_4$, but have little effect on PI(3,4,5)P$_3$ kinetics (*Figures 3E–F* and *7B–C*). This similitude in behavior supports an allosteric communication via R649 and D613/D615. Further, we find that the R682A and N684A mutants have lost most of their activity (*Table 3*; note that for N684A this is also the case for the CD-stable Ptase-C2 form), supporting a key role for these residues in substrate binding via 3 P and 4 P, respectively (*Figure 6A*). R691 links the Ptase core β-

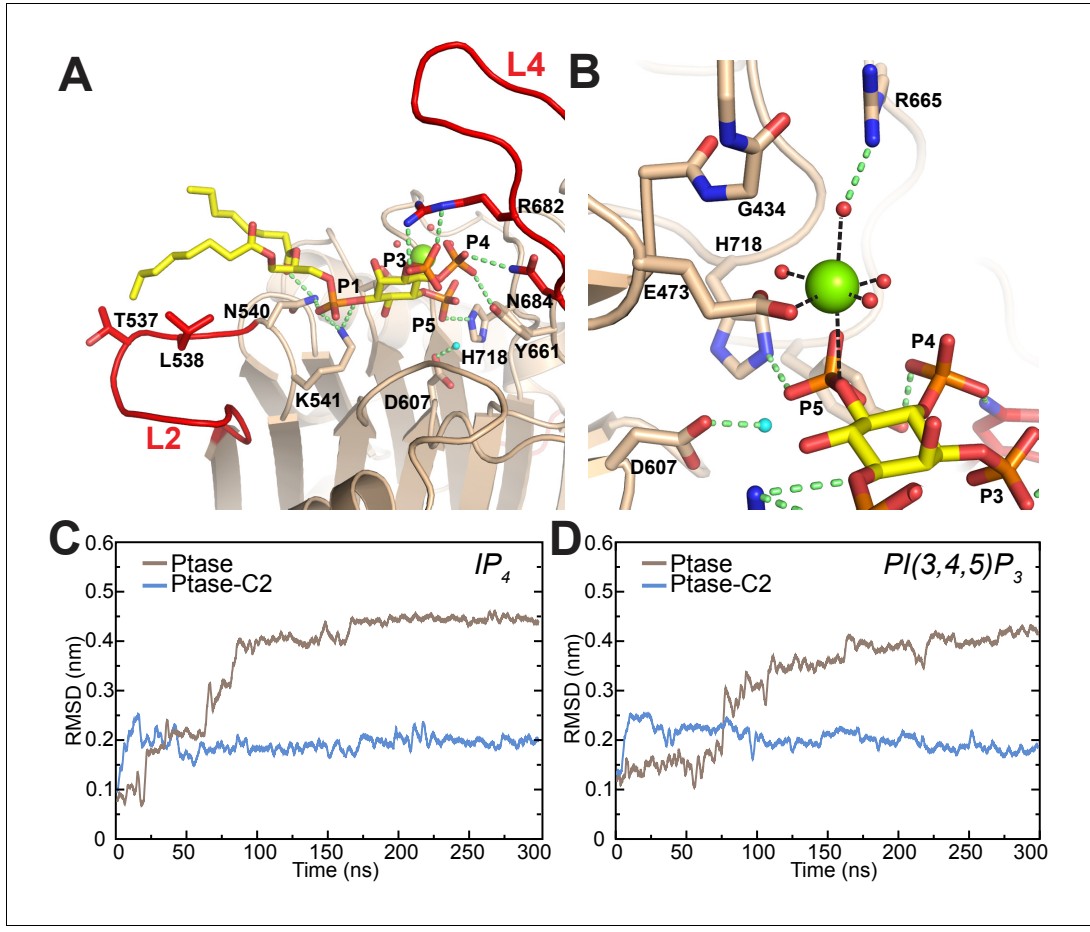

**Figure 6.** Model of substrate bound SHIP2. (**A**) The SHIP2 Ptase-PI(3,4,5)P$_3$-diC8 complex is modeled based on crystal structures of the homologue INPP5B crystal structure bound to PI(4)P (pdbs: 3MTC). L4 is in the 'in' conformation and R682 makes hydrogen bonds to the PI(3,4,5)P$_3$ 3-phosphate (P3). N540 and K541 interact with P1, N684 and Y661 with P4, H718 with P5 and L2 with PI(3,4,5)P$_3$ lipid chains. The attacking water, bound to D607, is colored light blue and the Mg$^{2+}$ ion is shown as green sphere. (**B**) Close-up of the Mg$^{2+}$ coordination. (**C–D**) Plotted are RMSDs of the substrate headgroup atoms compared to the starting positions during MD simulations of the SHIP2 Ptase or Ptase-C2 bound to IP$_4$ (**C**) or PI(3,4,5)P$_3$-diC8 (**D**). The simulations are started with substrate positions according to the model shown in panel (**A**).
The following figure supplement is available for figure 6:

**Figure supplement 1.** Structure of Ptase-C2 D607A, crystallized in presence of PI(3,4,5)P$_3$-diC8 and Mg$^{2+}$.

sandwich with helices α5–7 (**Figure 7A**) and its mutation inactivates Ptase-C2, which together with the changed CD spectra indicates a structural role for this residue. Unexpectedly, R665A mainly affects the Ptase domain, but has little effect on Ptase-C2 activity (**Figure 7D–E, Table 3**). According to our model, R665 stabilizes a water molecule in the coordination sphere of the Mg$^{2+}$ ion (**Figure 6B**). This fits with MD simulations showing C2-mediated stabilization of the substrate head-group (**Figure 6C–D**), apparently making the R665-mediated stabilization dispensable in presence of the C2 domain. Together, these results support an allosteric communication from the C2 domain via R649 and helices α5–7 to L4, which by changing their dynamic behavior affects enzyme kinetics with the IP$_4$ headgroup. In addition, they support an overall stabilization of the Ptase active site by the C2 domain allowing both substrates to bind in a catalytically productive mode.

Lastly, to probe how C2 interactions affect activity of full-length SHIP2 in a cellular setting we introduced the FLDD mutations or a C2 deletion (ΔC2) into the SHIP2 sequence and transiently expressed the full-length proteins in HEK293 cells (**Figure 8—figure supplement 1**). Expressing

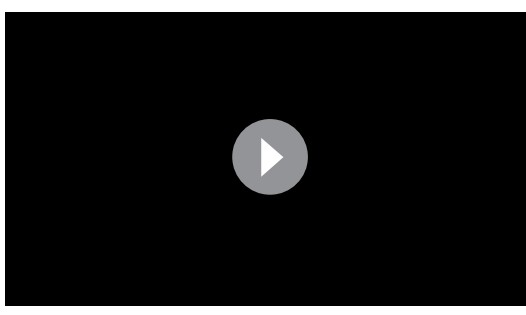

**Video 3.** Simulation of Ptase with substrate. The first 100 ns of the trajectory from the simulation of Ptase bound to PI(3,4,5)P$_3$-diC8 and Mg$^{2+}$ are shown.

wild-type SHIP2 causes a significant reduction in downstream Akt phosphorylation levels (**Figure 8**). Removing the C2 domain abolishes observed SHIP2 activity, whereas the FLDD mutations partially reduce activity of the wild-type protein. This confirms that C2 interactions observed in the crystal structure are important for SHIP2 activity in cells.

## Discussion

Of the enzymes controlling PI(3,4,5)P$_3$ levels (PI3K, PTEN and SHIP1/2), the SHIP enzymes lag far behind in terms of a detailed understanding of their functioning and regulation. Surprisingly, even though the Ptase domains of PTEN and SHIP2 bear no homology at the sequence or structural level, they conceptually share an important feature in that they are rigidly linked to a C2 domain (**Lee et al., 1999**). Despite the low homology between the PTEN and SHIP2 C2 domains (4% identity in a structural alignment) they use a common face for Ptase interactions. Indeed, a corresponding surface is also used by the C2 domain of PLCδ1 to form extensive interactions with its catalytic domain (**Essen et al., 1996**). It appears therefore, that the C2 domain presents an ideal scaffold to dock lipid modifying enzymes onto the membrane and via rigid interactions orient the catalytic domain for productive substrate attack. For SHIP2, we show that the C2 domain is essential for cellular function and the rigid interface enhances its efficiency (**Figure 8**). Assuming that the SHIP2 C2 domain engages the membrane via motifs known to bind lipids in other C2 domains (**Cho and Stahelin, 2006**) and taking into account several positively charged residues on the Ptase and C2 domains facing the membrane, we provide a model of SHIP2 Ptase-C2 docked to a lipid bilayer (**Figure 9A**). Although, we cannot rule out domain rearrangements between the Ptase and C2 domains upon lipid binding, the extensive domain interface between the two domains argues against large changes and suggests that the Ptase-C2 portion of SHIP2 represent the minimal fully active catalytic unit. Moreover, our model indicates that the rigid Ptase-C2 fragment can bind the membrane and engage substrate without the need of extensive extraction of the PI(3,4,5)P$_3$ substrate from the lipid bilayer. In SHIP2, PS binding occurs through both, the C2 and the Ptase domains and, as shown previously, interactions with the Ptase enhance SHIP2 activity (**Vandeput et al., 2006**). A similar scenario has indeed been observed for PTEN, where regions outside the C2 domain also contribute to membrane interactions and binding of acidic phospholipids to the N-terminus enhances activity (**Campbell et al., 2003**; **Walker et al., 2004**). The fact that the PS effect is larger when embedded in vesicles compared to soluble PS-diC8 (**Figure 4**) suggests that PS mediated SHIP2 localization and/or positioning on vesicles does contribute to activity on lipid membranes, however, binding via the Ptase appears sufficient for this effect.

Independent to a role in membrane binding, our data suggest that the C2 domain provides important allosteric interdomain effects to increase Ptase activity. In contrast to a traditional concept of allosteric signaling, effects by the C2 domain appear to be transmitted via changes in protein dynamics and active site stabilization rather than large conformational changes. Our data suggest that hydrophobic and polar regions at the Ptase-C2 domain interface differentially affect the lipid chains or head group of the PI(3,4,5)P$_3$ substrate. The fact that the R649A mutant, which presumably retains hydrophobic

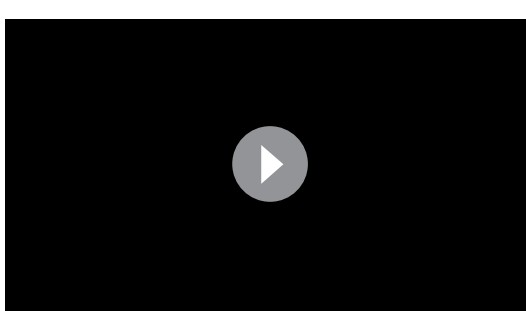

**Video 4.** Simulation of Ptase-C2 with substrate. The first 100 ns of the trajectory from the simulation of Ptase-C2 bound to PI(3,4,5)P$_3$-diC8 and Mg$^{2+}$ are shown.

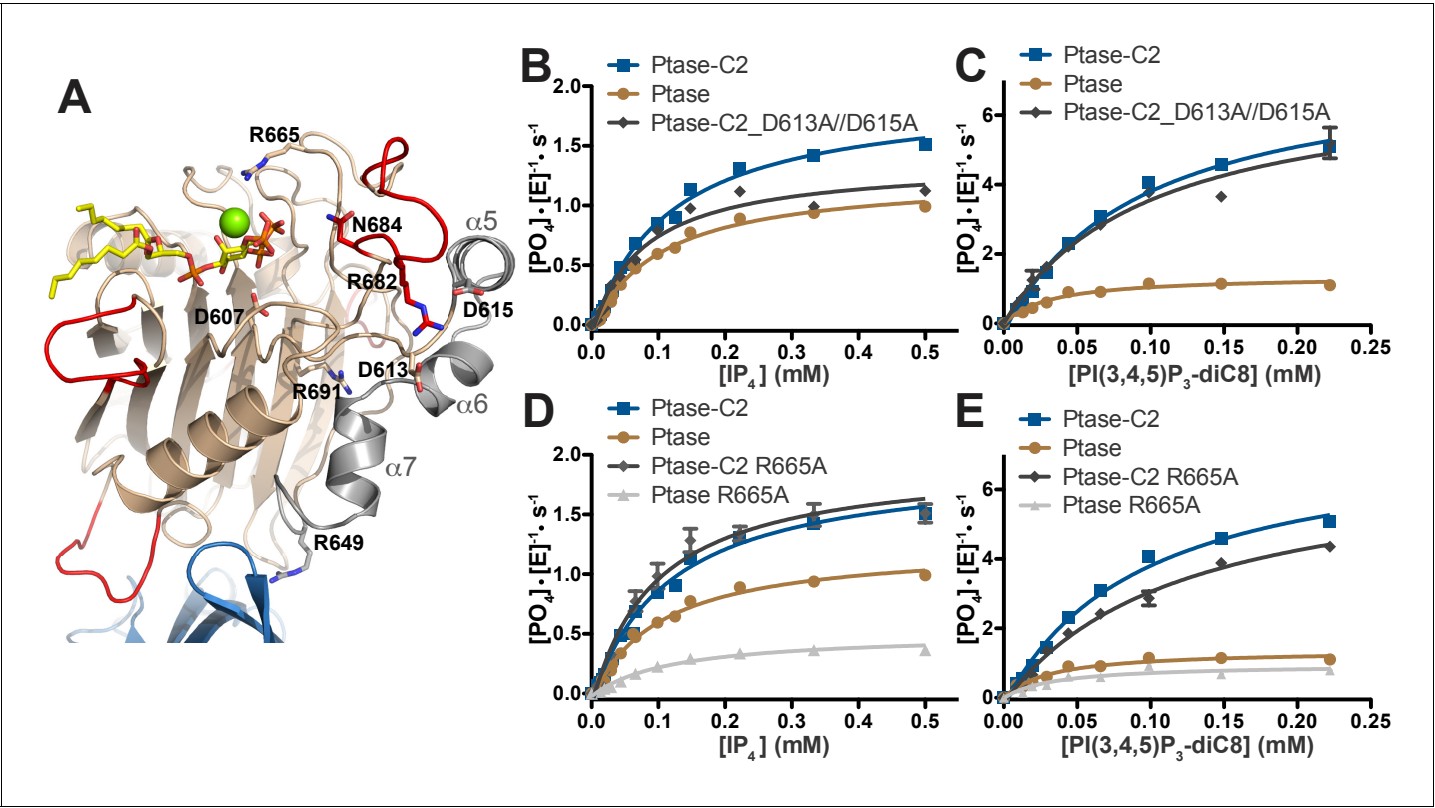

**Figure 7.** SHIP2 mutational analysis. (**A**) Residues mutated in the SHIP2 Ptase domain are shown as sticks and are labeled. (**B–E**) For mutants that display significant activity (D613A/D615A and R665A), substrate titration curves are shown. The enzyme concentration used in the shown plots is 400 nM (with IP$_4$) or 50 nM (with PI(3,4,5)P$_3$-diC8). Curves are fitted using the Michaelis-Menten equation and derived k$_{cat}$ and K$_M$ values are shown in *Table 3*. Error bars represent SEM from at least three measurements.

The following source data and figure supplement are available for figure 7:

**Source data 1.** Source data for plots in *Figure 7B–E*.

**Figure supplement 1.** Circular dichroism analysis of purified SHIP proteins.

Ptase/C2 interactions, only removes C2 effects on IP$_4$ but not PI(3,4,5)P$_3$ kinetics, indicates that R649 communicates with the PI(3,4,5)P$_3$ headgroup and hydrophobic interactions with PI(3,4,5)P$_3$ lipid chains. Since the PI(3,4,5)P$_3$ lipid chains are predicted to interact with L2 (*Figure 6A*), this suggests a communication between hydrophobic C2 interactions and L2.

Further, our study has identified a specific path of polar connections, linking the C2 domain to the Ptase active site. This 'polar' path leads via C2 interactions with R649 in the Ptase to helices α5–7 and at the other end of the helices to D613/D615 and R682 in L4. Mutation of R691 at the interface between helices α5–7 and the Ptase core inactivates SHIP likely due to structural destabilization (*Table 3*, *Figure 7—figure supplement 1B*), and similar inactivating mutations at this interface have been associated with opsismodysplasia (P659S and W688C) and protection from diabetes (L632I) (*Huber et al., 2013*; *Kagawa et al., 2005*). More specific disruption of this path, by either R649A or D613A/D615A mutations specifically removes C2 effects on IP4 (*Figures 3E–F* and *7B–C*). In a cellular setting this path could be relevant, firstly for SHIP2 activity on intracellular IP$_4$, whose cellular roles are only emerging for example in T-cell signaling (*Huang and Sauer, 2010*). Secondly, it is likely also important when the SHIP2 enzyme approaches the PI(3,4,5)P$_3$ substrate in membranes with the lipid chains buried in the lipid bilayer.

A critical observation in our study is that L4, proximal to the active site, can switch between a closed (L4-in) and an open (L4-out) conformation (*Figure 1D–F*) and our simulations indicate that the

**Table 3.** Enzymatic parameters are calculated by fitting the Michaelis-Menten equation to substrate titrations. Enzyme concentrations used for activity measurements were 400 nM, or if this caused saturated signals 50 nM (*). NA, not analyzed; ND, not determinable.

| | Ptase | | | | Ptase-C2 | | | |
| --- | --- | --- | --- | --- | --- | --- | --- | --- |
| | IP$_4$ | | PI(3,4,5)P$_3$ | | IP$_4$ | | PI(3,4,5)P$_3$ | |
| | $k_{cat}$ (s$^{-1}$) | $K_M$ ($\mu$M) | $k_{cat}$ (s$^{-1}$) | $K_M$ ($\mu$M) | $k_{cat}$ (s$^{-1}$) | $K_M$ ($\mu$M) | $k_{cat}$ (s$^{-1}$) | $K_M$ ($\mu$M) |
| R691A | NA | NA | NA | NA | ND | ND | ND | ND |
| D613A, D615A | NA | NA | NA | NA | 1.45 ± 0.05 | 76 ± 11 | 7.10 ± 0.57* | 100 ± 22* |
| R682A | ND | ND | ND* | ND* | 0.27 ± 0.04 | ND | 1.74 ± 0.23 | 93 ± 35 |
| N684A | ND | ND | ND | ND | ND | ND | ND | ND |
| R665A | 0.53 ± 0.03 | 119 ± 19 | 0.23 ± 0.01 | 43 ± 10 | 2.07 ± 0.09 | 94 ± 15 | 7.02 ± 0.37* | 119 ± 16* |
| D607A | NA | NA | NA | NA | ND | ND | ND | ND |

**Source data 1.** Source data for values shown in Table 3. All values are [PO$_4$] (in µM). Numbers (#$i$) above data indicate independent experiment number. Most kinetic parameters ($k_{cat}$, $K_M$) are extracted from curves shown in *Figure 6*, for which Source data are available with this figure. Below Source data are shown for cases where no curve is shown or where experiments were repeated at higher enzyme concentration (400 nM) to extract reliable kinetic parameters. Equation used to extract Vmax and $K_M$: Y = Bo + Vm*X/(X + $K_M$); Variables: Vmax, $K_M$, Bo = baseline. Software used: Graphpad Prism.

dynamics of this loop and the region capturing the loop in the open conformation is altered by the C2 domain (*Figure 5*, *Figure 5—figure supplement 1*). We propose a catalytic cycle for SHIP2 where efficient opening and closing of L4 is important for high turnover (*Figure 9B*, *Video 5*): To initiate the cycle, L4 needs to open to allow substrate entry to the active site. Initial interactions of the PI(3,4,5)P$_3$ head group include 4 P binding to N684, which prevents the formation of a hydrogen bond network seen in the R682 doubly bound L4-out conformation (*Figure 1F*). With this hydrogen network disrupted L4 is more mobile (see e.g. *Figure 1E*) facilitating the switch to L4-in. We propose closure of L4 over the substrate as an important second step in catalysis, allowing R682 interactions with the substrate 3 P. With the substrate tightly bound and correctly positioned, catalysis proceeds. Once 5 P cleavage has occurred we propose that L4 needs to open again for product release. In absence of 3 P and 5 P interactions, reestablishment of the hydrogen network as seen in the R682 doubly bound L4-out conformation that orients the N684 sidechain towards H674 (*Figure 1F*), might aid ejection of the product, hence closing the catalytic cycle.

It has been previously proposed that the 3 P specificity of SHIP2 is conferred by R682 (*Trésaugues et al., 2014*). Our model shows that R682 in the L4-in conformation is well placed to interact with the 3 P of PI(3,4,5)P3 (*Figure 6A*) and mutation of R682 confirms an important role for this residue in catalysis (*Table 3*). In other 5-Ptases the loop corresponding to L4 is significantly shorter and residues corresponding to R682 (K516 in INPP5B) and N684 (R518 in INPP5B) on L4 interact with the 4 P and belong to a set of residues defined as the P4 interacting motif (P4IM) (*Trésaugues et al., 2014*). Possibly, R682 in SHIP2 might also contribute to 4 P interactions, which could account for the observed PI(4,5)P$_2$ activity of SHIP2 (*Elong Edimo et al., 2016*; *Nakatsu et al., 2010*; *Taylor et al., 2000*), however, several reports demonstrate a clear preference for PI(3,4,5)P$_3$ (*Chi et al., 2004*; *Pesesse et al., 1998*; *Trésaugues et al., 2014*; *Vandeput et al., 2007*). Another difference in SHIP2 is the somewhat weaker Mg$^{2+}$ coordination compared to other 5-Ptases. An asparagine in INPP5B (N275) providing a Mg$^{2+}$coordination is replaced by a glycine in SHIP2 (G434) and the coordination is likely provided by a water molecule (*Figure 6B*). This water can be stabilized by R665, which appears important only in absence of active site stabilization by the C2 domain (*Figures 6C–D* and *7D–E*, *Table 3*).

Other head group interactions are conserved among 5-Ptases and the mechanism initiating 5 P attack is well described and shared among 5-Ptases as well as the AP endonucleases (*Dlakić, 2000*; *Whisstock et al., 2000*, *2002*). However, the exact mechanism of catalysis is still being debated. Different models have for AP nucleases proposed the involvement of either one or two static, or one moving metal ion (*Beernink et al., 2001*; *Mol et al., 2000*; *Oezguen et al., 2007*). A 2-metal hypothesis for AP nucleases is supported by a crystal structure of APE1, which has two Pb$^{2+}$ ions bound 5 Å

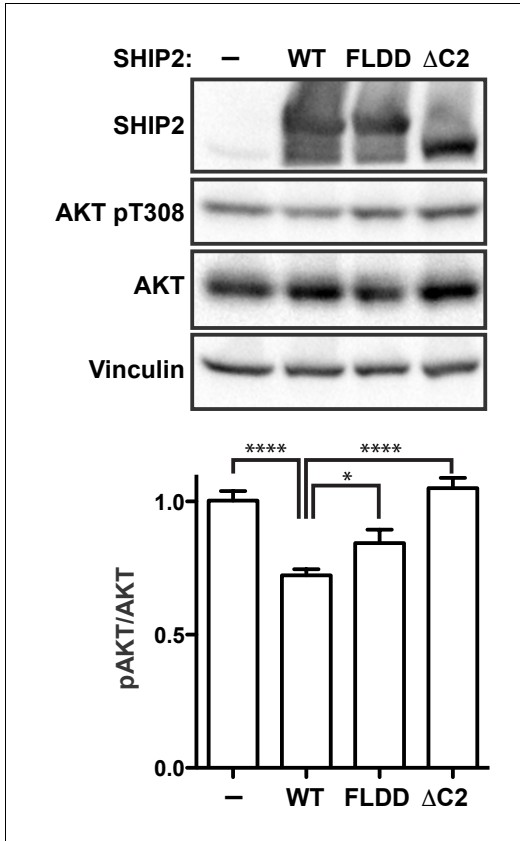

**Figure 8.** SHIP2 cellular activity. Full-length SHIP2 proteins were transiently expressed in HEK293 cells and resulting Akt-pT308 levels monitored. Shown are transfections of empty vector (-), wild type SHIP2 (WT), the FLDD mutant and C2 deleted SHIP2 (ΔC2). A typical blot is shown and quantifications of blots are averaged from eight independent experiments, each in triplicates (n = 24), with SEM's indicated. *p<0.05; ****p<0.0001 (unpaired Student t test).

The following source data and figure supplements are available for figure 8:

**Source data 1.** Source data for graph in *Figure 8*.

**Figure supplement 1.** Quantification of SHIP expression levels.

**Figure supplement 1—source data 1.** Source data for graph in *Figure 8—figure supplement 1*.

apart from each other, with one ion at the site seen in 5-Ptases (A-site) and the other at a more buried site (B-site) (*Beernink et al., 2001*). Similar to a recent structure of INPP5B (*Mills et al., 2016*), we observe in our Ptase-C2 D607A structure the $Mg^{2+}$ displaced on average by 1.3 Å from the A towards the B-site and a phosphate group shifted by 1.7 Å from the 5 P location towards the attacking water (*Figure 6—figure supplement 1B*). As suggested by *Mills et al. (2016)* this likely represents the post-cleavage position for $Mg^{2+}$ and the 5 P, since there is no longer space for the attacking water. In contrast, for OCRL a free phosphate is observed at the pre-cleaved 5 P site and the $Mg^{2+}$ at the A-site (*Trésaugues et al., 2014*), hence there appear to be subtle differences in the post-cleavage positioning of the $Mg^{2+}$ and phosphate for different 5-Ptases. We note that the movement of the $Mg^{2+}$ seen in INPP5B and SHIP2 from the A towards the B-site is opposite to the B to A-site movement proposed to occur during catalysis in AP nucleases (*Oezguen et al., 2007*).

SHIP1 shares an overall 45% sequence identity with SHIP2 (65% within the 5-Ptase and 43% in the C2 domain), suggesting that many of the mechanisms described here are conserved among the two enzymes. Residues at the Ptase-C2 domain interface are mostly conserved, suggesting that the interface as well as a possible communication to L2, which is fully conserved, are also present in SHIP1. The polar path is partially conserved with R649 replaced in SHIP1 by H632, D613/D615 are with D593 and E598 similar and R682 is substituted by K665. The equal length of L4 suggests that it will be able to sample a similar space and likely K665 provides the 3 P specificity in SHIP1.

In conclusion, our study provides important insights into how the C2 domain assists SHIP2 catalysis via a combination of membrane positioning, active site stabilization and allosteric signaling, which together is crucial for efficient cellular functioning (*Figure 8*). Since SHIP enzymes are highly implicated in disease, this will likely aid the design of novel strategies to target SHIP enzymes with high specificity.

## Materials and methods

### Protein expression and purification

The experimental details regarding the cloning, expression and purification of human SHIP2 Ptase-C2 have been described in (*Le Coq et al., 2016*). The Ptase was cloned, expressed and purified following the same protocol, whereas the isolated C2 could not be purified due to insolubility of the protein.



**Figure 9.** Model of SHIP2 catalytic cycle. (**A**) Model of SHIP2 Ptase-C2 docked to the membrane and with PI(3,4,5) $P_3$-Mg$^{2+}$ bound to the active site. Basic and hydrophobic side chains of K531, I534, L538, K568, R571, R581, R588 and L590 in the Ptase and R762, K764, K779, K826 and R859 in the C2 domain (shown as sticks) are modelled to contribute to membrane interactions. L2, containing I534 and L538 is modeled to penetrate the lipid bilayer. The

*Figure 9 continued on next page*

*Figure 9 continued*

C2-PS interaction is based on PDB entry 1DSY and PI(3,4,5)P₃ interactions on the model shown in *Figure 6A*. (B) The catalytic cycle. Top: Docking of Ptase-C2 to the membrane orients via the rigid domain interface the Ptase active site towards its membrane substrate. Step 1: L4 is 'out' to allow entry of the PI(3,4,5)P₃ headgroup. Step 2: Initial engagement of the substrate 4 P with N684 destabilizes the doubly bound L4-out conformation and initiates the switch to L4-in, which allows R682 interactions with the substrate 3 P. L2 opens and penetrates the membrane to interact with PI(3,4,5)P₃ lipid chains. Step 3: With the substrate correctly positioned, catalysis proceeds and the cleaved 5 P and the Mg²⁺ion move towards the metal B-site (see text). Step 4: Product interactions are weakened by L4 moving 'out', releasing R682 from the product 3 P. Further, N684 releases 4 P interactions by switching to bind H674 (see *Figure 1F*). Step 5: The weakly bound product is released and a new substrate can be engaged.

## Crystallization, data collection

The crystallization conditions for the Ptase-C2 have been previously described (*Le Coq et al., 2016*). Briefly, crystals were grown by mixing an equal volume of the protein with the precipitant solution (100 mM Bis-Tris propane, pH 7.0, 0.2 M NaNO₃, 20% (w/v) PEG3350, 0.025% (v/v) CH₂Cl₂). Crystals were cryoprotected in the precipitant solution with 25% ethylene glycol and then flash-frozen. SHIP2 Ptase-C2 D607A (5.5 mg/mL) in the presence of PI(3,4,5)P₃-diC8 and Mg²⁺ (0.450 mM and 0.750 mM respectively) and SHIP2 Ptase-C2 FLDD (5.32 mg/mL) crystallization conditions were initially rescreened. For SHIP2 Ptase-C2 D607A mixed with PI(3,4,5)P₃diC8 and Mg²⁺ conditions were optimized to 0.1 M Bis-Tris propane, pH 7.0, 0.4 M KSCN, 20% (w/v) PEG3350 for SHIP2 Ptase-C2 D607A mixed with PI(3,4,5)P₃diC8 and Mg²⁺. The crystals were flash-frozen in precipitant with additional 25% ethylene glycol, 0.220 mM of PI(3,4,5)P₃diC8 and 0.750 mM of Mg²⁺. For SHIP2 Ptase-C2 FLDD crystals were grown in 50 mM Hepes pH 7.1, 17.5% PEG 1000 and cryopreserved in additionally 25% ethylene glycol (I₂ crystals) or 0.1 M PCB buffer pH 7 (sodium propionate, sodium cacodylate, and BIS-TRIS propane; 2:1:2 molar ratio), 25% PEG 1500 and cryopreserved in additionally 25% ethylene glycol (P2₁ crystals).

## Structure determination, Refinement, and Analysis

Crystallographic data for all protein crystals were measured at 100K, on beamlines ID23-2 (λ = 0.873 Å) and ID29 (λ = 0.976 Å) at European Synchrotron Radiation Facility (ESRF, Grenoble) and BL13-XALOC-BL13 (ALBA, Barcelona, for preliminary data collection) using a Pilatus 6M detector. XDS (*Kabsch, 2010*) was used for indexing and integration and AIMLESS (*Evans and Murshudov, 2013*) for scaling all diffraction data. The best dataset for SHIP2 Ptase-C2 WT diffracted to a resolution of 1.96 Å and reveals a P2₁2₁2₁ space group with unit cell parameters a = 136.04 Å, b = 175.84 Å, c = 176.89 Å. The Matthews coefficient is 2.54 Å³.Da⁻¹ corresponding to 8 molecules in the asymmetric unit (asu) with a solvent content of 51.7% (*Le Coq et al., 2016*). The crystal structure of *human* SHIP2 Ptase model (PDB: 3NR8) was used as search probe for molecular replacement (MR) using Phaser (*McCoy et al., 2007*), which identified 8 clear solutions for the Ptase domain. MR using various available C2 structures as search probes failed. C2 domains where therefore built using ARP/wARP (*Langer et al., 2008*) and manual building with COOT (*Emsley and Cowtan, 2004*; *Emsley et al., 2010*). Translation/Libration/Screw (TLS) and Maximum likelihood of restrained refinement were performed using REFMAC5 (*Murshudov et al., 1997*). Final R-factors for SHIP2 Ptase-C2 WT are 18.0/20.8 (Rwork/Rfree). SHIP2 Ptase-C2 D607A crystallized in the same space group as wild type SHIP2 Ptase-C2, but Phaser (*McCoy et al., 2007*) was used to generate the initial model

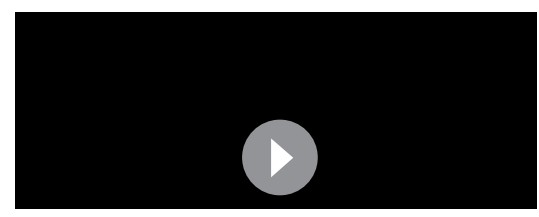

**Video 5.** The SHIP2 catalytic cycle. The conformational changes between L4-out doubly bound, L4-out singly bound and L4-in conformations are shown as morphs between observed crystal structures and assembled in sequence according to the model of the catalytic cycle explained in *Figure 9B*.

since it produced better initial R values than directly using the SHIP2 Ptase-C2 WT model. Refinement was performed as described above. A $Mg^{2+}$ and a $PO_4^{3-}$ were placed in the active site of 6 molecules in the asu (monomers C-H). SHIP2 Ptase-C2 FLDD crystallized in two different space groups, one in $P2_1$ (a = 44.03, b = 81.12, c = 128.90, $\beta$ = 92.85) with 2 molecules per asu and the other in $I_2$ (a = 43.74, b = 73.43, c = 158.0, $\beta$ = 90.7) with 1 molecule per asu. The Ptase-C2 WT model was used in Phaser (*McCoy et al., 2007*) to provide initial phases. Initial rebuilding was performed with ARP/wARP (*Langer et al., 2008*) and further refinement was performed as described above. All data processing and structure refinement statistics are summarized in *Table 1*. All residues in the final models lay within the favored region of the Ramachandran plot with the exception of the model of SHIP2 Ptase-C2_D607A which has 0.2% Ramachandran outliers.

## CD spectroscopy

Circular dichroism was used to assess and compare the overall protein fold of the different constructs. Far-UV CD spectra between 250 and 200 nm of protein at 0.25 mg/mL were run at 20°C for each sample.

## Thermal melting

Tm's were measured using a ThermoFluor assay. Protein at 1–10 µM was mixed with SYPRO Orange and subjected to a temperature gradient of 0.5°/min from 20 to 95°C and fluorescence recorded at 570 nm.

## Lipid binding

For the PLO assay dipalmitoyl phosphatidylserine or dipalmitoyl phosphatidylcholine (Echelon Biosciences, Salt Lake City, UT) were spotted on a nitrocellulose membrane, dried and blocked with 5% skimmed milk. The membranes were then incubated for 1 hr with 2.5 µg/ml of GST fused SHIP Ptase or Ptase-C2 in blocking solution in presence or absence of 1 mM $CaCl_2$, washed and developed using an HRP conjugated anti-GST antibody.

For SPR measurements vesicles containing 30% PS (16:0; Echelon Biosciences) and 70% PC (from chicken egg; Avanti Polar Lipids, Alabaster, AL) or 30% PC (16:0, Avanti Polar Lipids) and 70% PC (chicken egg) were prepared at a final total lipid concentration of 1.5 mM. Organic solvent was removed by rotary evaporation for 1.5 hr at 45°C. The lipid film was resuspended in SPR running buffer (20 mM HEPES, pH 7.5, 150 mM NaCl, 1 mM TCEP) and subjected to six cycles of freeze-thaw and passed 15 times through a membrane with 100 nm pore size, using a mini extruder (Avanti Polar Lipids). SPR experiments were performed on a Biacore X100 instrument (GE Healthcare, Chicago, IL). A L1 sensor chip was washed with two 1 min injections of a 2:3 (v/v) isopropanol: 50 mM NaOH solution at a flow rate of 10 µl/min and then coated by injecting PS containing vesicles in the active flow cell (Fc2) and PC vesicles in the reference flow cell (Fc1), for 15 min at a flow rate of 2 µl/min, followed by two 1 min injections of 10 mM NaOH solution at 10 µl/min to remove loosely bound vesicles and for stabilization. SHIP2 Ptase or Ptase-C2 proteins were injected for 2 min at a flow rate of 30 µl/min in running buffer. SPR responses plotted in *Figure 2C*-insert and *Figure 2D* correspond to response units (RUs) at 10 s after injection, when a relatively constant steady state binding phase is reached (dashed line in *Figure 2C*). Injections in presence of $Ca^{2+}$ were performed with 0.5 mM $CaCl_2$. For all data shown in *Figure 2C–D*, buffer injections (with or without calcium, accordingly) are subtracted.

## Enzyme kinetics

Activity measurements were performed using a Malachite Green phosphatase activity assay. Proteins were incubated for 2 min at 23°C with the substrate in a reaction buffer containing 20 mM Hepes, 150 mM NaCl, 2 mM $MgCl_2$, 1 mM TCEP, pH 7 in a total volume of 25 µL. The reaction was quenched by addition of 5 µL of 0.5 M EDTA, pH 8. Subsequently, 25 µL of the reaction was mixed with 100 µL of Malachite Green solution (Echelon Biosciences) and left to incubate for 15 min at room temperature and the optical density was measured at 620 nm. Each data point is measured at least in triplicates. For activity assays shown in *Figures 4A–B* 100 µM soluble PS-diC8 or PC-diC8 were included.

For activity measurements with vesicles two sets of PI(3,4,5)P$_3$ vesicles were prepared at a total lipid concentration of 4.5 mM, containing either 10% (mol/mol) PI(3,4,5)P$_3$ (16:0; Echelon Biosciences), 30% (mol/mol) PS (16:0, Echelon Biosciences) and 60% (mol/mol) PC (from chicken egg, Avanti Polar Lipids) or for control vesicles PS was replaced by 30% (mol/mol) PC (16:0, Avanti Polar Lipids). The organic solvent was removed by rotary evaporation for 90 min at 45°C. The lipid film lipid was subsequently resuspended in 20 mM HEPES, 150 mM NaCl, 1 mM TCEP, pH 7 and sonicated for 35 min. To calculate the dilution of vesicles in experiments shown in *Figure 4C–D*, only half of the total PI(3,4,5)P$_3$-diC16 concentration, present in the outer leaflet of vesicles, was considered.

## Cell assay

HEK293 cells were maintained in Dulbecco's modified Eagle's medium (DMEM) containing 10% fetal bovine serum (FBS) and supplemented with antibiotics. The cells were transiently transfected with the empty vector (pOPINJ) or vector expressing GST fused full-length SHIP2 (WT, FLDD or ΔC2). The transient transfection was performed using polyethylenimine (*Hsu and Uludağ, 2012*). After 48 hr in DMEM with 10% FBS the cells were harvested and levels of total Akt, phospho-T308 of Akt (Cell signaling, Danvers, MA, AB_2255933, AB_329827) and SHIP2 (Abcam, UK, AB_2686895) expression levels were assessed by immunoblotting. Quantifications were performed with ImageJ on eight independent experiments, each in triplicates (n = 24). Standard errors in *Figure 8* are calculated using values of pAkt/Akt ratios and for *Figure 8—figure supplement 1* SHIP2/Vinculin (AB_477629) ratios. The significance is calculated using a two-tailed unpaired Student t test assuming Gaussian distribution.

## MD simulations

For apo simulations, the initial model was based on molecule B of the SHIP Ptase-C2 WT structure, which has a L4-out conformation. Missing residues in the linker (residues 731–746) were modelled with ModLoop (*Fiser et al., 2000*) and energy minimized. For substrate bound models L4 was replaced with an L4-in conformation from the SHIP2 Ptase (3NR8). The position of PI(3,4,5)P$_3$, Mg$^{2+}$- and catalytic waters were based on the INPP5B crystal structure bound to PI(3,4)P$_2$ (3MTC, adding the 5 P and 3 P) and based on AP endonucleases (*Whisstock et al., 2002*). Lipid parameters for IP$_4$ and PI(3,4,5)P$_3$ and a net charge of −5 for protein bound IP$_4$/PI(3,4,5)P$_3$ were based on previous studies (*Rosen et al., 2011*; *Slochower et al., 2013*). For Ptase simulations the C2 and linker residues (733-874) were omitted. Hydrogens were added using the H++ server (*Gordon et al., 2005*). The systems were solvated in a cubic simulation cell of 80 Å edge length, constructed from an equilibrated box of TIP3P water molecules (*Mahoney and Jorgensen, 2000*). The size of the simulation cell was chosen such that a distance of at least 10 Å between the surface of the protein and the cell boundaries was maintained. The systems were neutralized with Na$^+$/Cl$^-$. All calculations were carried out with periodic boundary conditions. All complexes were minimized in three stages. In the first stage, the water molecules and the ions were energy minimized, while harmonic restraints of 1000 kJ/(mol·nm$^2$) were applied to the protein and the lipid; 500 steps of steepest descent and 500 steps of conjugate gradient minimization were carried out. In the second stage the lipid harmonic restrains were removed and another 500 steps of minimization were carried out. In the last stage, all restrains were removed and the entire system was minimized for 2000 steps. The resulting configurations were used for MD production: the lipid systems were simulated for a total of 300 ns each, and the non-lipid simulations for 3 × 1.5 μs each (three independent simulations for each system). The first 50 ns of each simulation were considered equilibration time and omitted from the analysis. All minimization and MD production were performed using version 4.6 of the molecular simulations package GROMACS (*Hess et al., 2008*; *Van Der Spoel et al., 2005*) with the AMBER99SB*-ILDN force field (*Lindorff-Larsen et al., 2010*) with the dihedral corrections of Best Hummer (*Best and Hummer, 2009*). The isobaric-isothermal (NPT) ensemble was employed for all MD calculations at 300 K with a velocity-rescale thermostat (*Bussi et al., 2007*) and a time step of 2 fs.

## RMSF and RMSD analysis

RMSF and RMSD were calculated on the C$_\alpha$ of the 5-Ptase domain from the total 4.35 μs MD simulations, using the g_rmsf and g_rms tools from the GROMACS package. RMSDs were calculated per residue, using the L4-out structure as reference, and averaged over time. For plots shown in

*Figure 5—figure supplement 1A–F*, RMSF and RMSD were calculated separately over snapshots, in which R682 was unbound, singly bound to either D613 or D615, or doubly bound to D613 and D615. Bonds were considered for minimal R682-D613/D615 distances below 4 Å (excluding hydrogens), hence including hydrogen bond and electrostatic bonding character.

### Principal component analysis

PCA was performed on all $C_\alpha$ atoms, applying g_covar and g_anaeig tools from GROMACS package on the full MD trajectories, to calculate, diagonalize and analyze the covariance matrix respectively. PCA identifies collective motions associated with the largest variance and the lowest frequencies shown in the first few eigenvectors, thus, we analyzed the first eight PC of each simulation.

### Cluster analysis

Cluster analysis was performed on all non-hydrogen atoms using g_cluster with the gromos algorithm (*Daura et al., 1999*) and a RMSD cutoff of 0.15 nm. The structure with the smallest average distance to the others was written as the representative structure in each cluster.

## Acknowledgements

We thank José Terrón Bautista for help with MD analysis. We thank the ESRF and ALBA for providing the synchrotron-radiation facilities and the staff for their assistance in the data collection. We are grateful to the Barcelona Supercomputing Centre and National Supercomputing Centre (BSC-CNS) for allocating computer time to run the reported simulations. The work was supported by the Spanish Ministry of Economy, Industry and Competitiveness (MEIC) Grants BFU2010-15923 (DL) and MEIC Project Retos BFU2016-77665-R co-funded by the European Regional Development Fund (ERDF) (DL), the Comunidad Autónoma de Madrid Grant S2010/BMD-2457 (DL), and by the National Cancer Research Centre. DL is also a recipient of awards from the Volkswagen Foundation (Az: 86 416–1) and Worldwide Cancer Research (15-1177).

## Additional information

### Funding

| Funder | Grant reference number | Author |
| --- | --- | --- |
| Ministerio de Economía, Industria y Competitividad | BFU2010-15923 | Daniel Lietha |
| Comunidad de Madrid | S2010/BMD-2457 | Daniel Lietha |
| Ministerio de Economía, Industria y Competitividad | Project Retos BFU2016-77665-R | Daniel Lietha |
| European Regional Development Fund | Project Retos BFU2016-77665-R | Daniel Lietha |

The funders had no role in study design, data collection and interpretation, or the decision to submit the work for publication.

### Author contributions

JLC, Conceptualization, Data curation, Formal analysis, Validation, Investigation, Visualization, Methodology, Writing—original draft, Writing—review and editing; MC-A, Data curation, Formal analysis, Investigation, Visualization, Writing—review and editing; JVV, Investigation, Methodology, Writing—review and editing; CMS, Investigation, Writing—review and editing; LHG, Formal analysis, Investigation; RC-O, Formal analysis, Investigation, Writing—review and editing; ND, Formal analysis, Supervision, Writing—review and editing; DL, Conceptualization, Data curation, Formal analysis, Supervision, Funding acquisition, Investigation, Visualization, Writing—original draft, Project administration, Writing—review and editing

## Author ORCIDs

Johanne Le Coq, http://orcid.org/0000-0002-5309-3795

José Vicente Velázquez, http://orcid.org/0000-0002-6282-5724

Daniel Lietha, http://orcid.org/0000-0002-6133-6486

## Additional files

**Supplementary files**

• Supplemental file 1. Thermal melting (Tm) of SHIP2 WT and mutants. Tm values are determined by thermofluor. NA, not analyzed; WS, weak signal.

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
