## [Decision Letter]

Thank you for submitting your article "Structural basis for interdomain communication in SHIP2 providing high phosphatase activity" for consideration by *eLife*. Your article has been favorably evaluated by John Kuriyan (Senior Editor) and three reviewers, one of whom is a member of our Board of Reviewing Editors. The following individual involved in review of your submission has agreed to reveal their identity: Evzen Boura (Reviewer #3).

The reviewers have discussed the reviews with one another and the Reviewing Editor has drafted this decision to help you prepare a revised submission.

Previous work has established the structure of the SHIP2 phosphatase domain and its relationship with other members of the 5-phosphatase family. This manuscript has established the structure of a pptase-C2 domain construct and examined the role of the C2 domain in catalysis. The current study represents a significant advance in understanding SHIP2. The structure shows that the C2 domain makes an extensive interaction surface with the phosphatase domain. A similar interaction of the C2 domain with the catalytic domain has been observed in other lipid-modifying enzymes, such as PLCδ1 and PTEN. The manuscript establishes that the C2 domain acts as an allosteric activator by stabilising the pptase domain and contributing to membrane binding. The manuscript outlines two distinct pathways of stabilisation contributing to the effect of the C2 domain, a polar pathway manifest on activity with the soluble substrate IP4 and a hydrophobic pathway that specifically affects interactions with the acyl chains of PIP3.

The combination of structures and molecular dynamics provides a compelling model for the catalytic cycle of SHIP2 encountering substrate in lipid membranes, with the L4 loop changing conformation between an initial encounter state and a state in which substrate is engaged in the active site. This is an important contribution to our understanding of this family of enzymes. It also provides a detailed picture of how the C2 domain of SHIP2 participates in membrane binding and catalysis.

Several points made by the reviewers (see below) should be taken into account in the revised manuscript. The reviewers have found the allosteric mechanism for the effect of the C2 domain based on crystallography and MD are convincing, but it would be helpful if you could provide a movie that portrays this mechanism. In the revision, you should take account of the suggestions by the reviewers, in particular, in the discussion of an effect on k_cat_ and K_M_ as being indicative of allostery (reviewer 1).

An important concern that has been raised by the reviewers is that the lipid blot demonstration of lipid affinity is not adequate for the work. These blots are subject to a range of artifacts. It would be better if you could measure the affinity using lipid vesicles and the FRET assay suggested by reviewer 3. This assay has been described by a number of workers (see, for example, PMID 22949682). Also, the methods should be revised to make clear exactly what lipids are being used and how the lipids are prepared for the assay. While the reviewers believe that assays with substrate in lipid vesicles would be an informative addition to the manuscript, it may be that this is beyond the scope of the present work, and this should not be regarded as essential.

In an earlier study of several phosphatases (Nordlund and colleagues), the L4 loop of SHIP2 was referred to as the P4-interacting motif (P4IM). It would be helpful for readers familiar with the previous study if the current manuscript also at least mentions this nomenclature.

It is claimed at the end of the manuscript that the cellular assays presented show that the C2 domain has an additional role, likely in membrane localization. This statement should be clarified, or if the statement is not based on the results, this should be explained.

The manuscript claims that "both, the Ptase and C2 domains of SHIP2 bind PS in a Ca^2+^ independent manner" This is misleading. The C2 domain on its own was never tested. It could be that the interaction depends entirely on the pptase domain, but that the pptase is only oriented and stabilised by the C2 domain. The statement consistent with the results would be "both the Ptase only and the Ptase-C2 constructs bind PS in a Ca^2+^-independent manner."

Common membrane-binding mechanisms are asp/glu residues binding Ca^2+^, which in turn binds to membranes or lys/arg residues that bind directly to the negatively charged lipids. Figure 2 shows that the SHP2 residues in the key positions are ser or asp/glu residues, but Figure 2 shows that Ca^2+^ does not affect binding. It would be helpful if you briefly discuss this point.

For assistance to color blind readers, it would be helpful if the explanatory keys in Figure 6 are placed next to the curve that they describe, or that there is a line between the curve and the key. There are shapes for markers on the curves in the figure, but these are typically small in the final version and may not be clear on their own.

In Figure 7, since the ratio pAkt/Akt is not the directly observed quantity, the manuscript should make clear how the errors in this ratio were determined. Error propagation in the ratio based on the standard deviations of the two observed quantities should have been carried out. The reviewers are concerned that differences in activity are complicated by differences in expression. To allay these concerns, the statistical methods should be described in the Materials and methods.

The WT X-ray dataset appears to be the same data set that was described in the earlier crystallisation report, however, the I/σ and highest resolution shell limits are not the same. It appears that the data have been reprocessed, and this is not surprising. However, it should be made clear how the resolution limits were chosen. Since the I/σ for the highest-resolution shells differ for the various datasets, it does not seem that this was the decisive criterion. The correlation coefficients cc1/2 and cc* have been proposed as a criterion to help make resolution cutoff in a statistically reliable manner. The table should also include cc1/2.

Reviewer #1:

Previous work has established the structure of the SHIP2 phosphatase domain and its relationship with other members of the 5-phosphatase family. The current study by Lietha and colleagues has established the structure of pptase-C2 domain construct and examined the role of the C2 domain in catalysis. A previous report by the same group described crystallisation and diffraction of the pptase-C2 construct and demonstrated its enzymatic activity. The current study represents a significant advance in understanding this enzyme. The structure shows that the C2 domain makes an extensive interaction surface with the phosphatase domain. A similar interaction of the C2 domain with the catalytic domain has been observed in other lipid-modifying enzymes, such as PLCδ1 and PTEN. The authors establish that the C2 domain acts as an allosteric activator by stabilising the pptase domain and contributing to membrane binding. The authors are suggesting two distinct pathways of stabilisation contributing to the effect of the C2 domain, a polar pathway manifest on activity with the soluble substrate IP4 and a hydrophobic pathway that specifically affects interactions with the acyl chains of PIP3.

The study provides a compelling model for the catalytic cycle of SHIP2 encountering substrate in lipid membranes, with the L4 loop changing conformation between an initial encounter state and a state in which substrate is engaged in the active site. This is an important contribution to our understanding of this family of enzymes. It also provides a detailed picture of how the C2 domain of SHIP2 participates in membrane binding and catalysis. This is particularly useful, since a similar catalytic domain/C2 module is present in other lipid-modifying enzymes.

In a previous study by Nordlund and colleagues, structures of SHIP2, OCRL, and INPP5B pptase domains were determined and compared. In this previous study, it was shown that a motif referred to as the P4-interacting motif (P4IM) interacts with 3- and 4-phosphates of the substrates. In the manuscript of Lietha and colleagues, the same feature is referred to as loop L4. To simplify things for the readers, it would be helpful if the authors of the current manuscript use the P4IM notation or at least point out that this is contained in L4. The presence of the C2 domain results in a large change in the conformation of the L4 loop, and the authors propose that this change likely constitutes an important component of the catalytic cycle of SHIP2 on membranes.

In the first paragraph of the subsection “The C2 domain affects SHIP2 activity” it states that changing k_cat_ without much change in K_M_ suggests that the activation is allosteric. The most common definition of allosteric is an effect on activity due to interaction outside the catalytic site. This can be an effect on K_M_ (K-type allostery), or it can be an effect on V_max_ (V-type), or come combination of these. However, simply an effect on k_cat_ does not suggest an influence directed from outside the active site. A k_cat_-only effect could be obtained by forming additional interactions with the transition state, but not with the substrate, but this would not be an allosteric effect. What the authors mean is that if the C2 domain has any influence at all, it must be allosteric, since the C2 domain is not near the active site. This should be restated. There is also a grammatic problem with the sentence in its current form (should be "suggests" instead of "suggesting").

In the last paragraph of the subsection “The C2 domain affects SHIP2 activity”, describing Figure 3, it is not clear what the authors have done in the experiments that they describe as adding "PC" and "PS". What exactly are the lipids that they have used? If they are synthetic, they should state what the acyl chains are. If they are from a natural source, they should quote the dominant component of the source. It is particularly important the length and saturation of the acyl chains. Also, there is no description of how the lipid substrates were presented. Were they prepared as unilamellar vesicles? Were the multilammelar vesicles? Were they SUVS? LUVS? GUVS? The reason that this is important is that the authors are using a di-C8 PIP3 lipid that is soluble on its own, so it is not clear that this lipid would effectively and completely partition into membranes of lipid vesicles. If the lipid substrate is both in the bulk phase and in the membranes, it makes interpretation of the results more complex. Ideally, it would be helpful if the authors had used long-chain PIP3 incorporated into unilamellar vesicles for at least some of their assays, in order to strengthen their conclusions for the behavior of SHIP2 in cells.

In the aforementioned paragraph, it is proposed that PS orients the enzyme on PIP3-containing membranes. This is an interesting proposition that could be tested if the substrate were PIP3 in vesicles, however, the authors are using di-C8 PIP3, which is soluble and not necessarily entirely present in membranes. It is hard to assess what is happening in Figure 3, which is intended to show the effect of PS. In Figure 3, Both the pptase and pptase-C2 constructs are activated by PS, so there is little indication in this that the C2 domain is contributing to the activation by PS. Figure 2 suggests that the pptase-C2 construct binds better to PS than the pptase domain alone. However, from 3G, this does not translate to PS-enhanced activity. This is perhaps not surprising, since the di-C8 PIP3 substrate is soluble and perhaps not fully incorporated into the undescribed PS-containing membranes. This experiment would be far more informative if the study were using PIP3/PC/PS unilamellar vesicles with long-chain fatty acids on all of the lipids. I would not regard these vesicle assays as essential for the paper, but they would enhance the study.

In the last paragraph of the subsection “Probing the communication path by mutagenesis”, the authors state that their cellular results confirm that C2 interactions observed in the crystal structure are important for SHIP2 activity in cells and that the C2 domain has an additional role, likely in membrane localization. It is not clear how these results show that the C2 domain has an additional role in cells. This should be clarified or the statement removed.

Reviewer #2:

These authors examine interdomain regulation for the SHP2 phosphatase. In particular, how does the C2 domain affect the phosphatase domain? They report a number of x-ray crystal structures as well as circular dichroism spectra and phosphatase kinetics. Finally, they refine the results with molecular dynamics simulations.

They find two conformations: loop L4 in and out. They propose that the C2 domain influences this conformational change. They do a good job with a schematic for their mechanism in Figure 8. This study includes a very large number of experiments and addresses an important question. However, there are a number of points that could be improved. My suggestions are as follow:

1) Figure 2. I found this figure confusing. The most common mechanisms of binding negatively charged lipids are asp/glu residues binding Ca^2+^ or lys/arg residues that directly to the negatively charged lipids. Figure 2 shows that the SHP2 residues in the key positions are ser or asp/glu residues. There are no positively charged amino acids so one would predict a role for Ca^2+^. But Figure 2 shows that Ca^2+^ does not affect binding. This says to me that Figure 2 is misleading. The binding site must be elsewhere in the C2 domain.

2) Figure 6—figure supplement 1. I am partly color blind. It would be convenient if the figure legend identified the mutants that have different spectra.

3) Figure 7. This experiment examines two mutants in HEK293 cells. This figure surprised me in two ways. The recombinant SHIP2 is a dark band. The endogenous SHIP2 is not detected in the control lane so the SHIP2 is grossly over expressed but there is only a 40% decrease in pAkt. Second, the deletion of the C2 domain clearly reduces activity, but the FLDD mutation has only a minute affect, which they say is statistically significant. They do not explain the statistical analysis. How was the error for pAkt propagated with the error in Akt? I am just not convinced that the small effect of the FLDD mutation is real and not due to error or differences in expression. They have a supplemental figure for expression. The FLDD is slightly lower but they say that it is not statistically significant. I need a little bit more convincing, and the statistics should be in Materials and methods.

Figure 8 liked this schematic. Very helpful.

Reviewer #3:

Le Coq et al. Report the crystal structure of a SHIP2 construct that, unlike previously reported structures, contains the membrane targeting C2 domain. The reported structural analysis together with molecular dynamics simulations reveals how the C2 domain (and thus membrane binding) allosterically controls the enzymatic activity of SHIP2. The findings are important, most of experiments are well done, manuscript is well written and the reasoning is sound and square. The manuscript clearly deserves to be published in *eLife*. However, there is one major issue (and several small technical issues) that should be improved before final acceptance. The additional control experiments should not take more than a day or two of a skilled experimentalist.

1) A protein lipid overlay assay is a useful screening technique, however, no conclusions should be derived from this method as it is notoriously known to be artifact prone. The authors need to verify their results using liposomes with/out PS and with/out Ca^2+^. Many reliable standard assays exist. Since SHIP2 contains trypthophan residues I suggest to use a FRET based assay (W residues as donors and dansyl-PE as acceptors) to measure binding.

2) Biochemical (enzymatic) analysis is done using soluble substrates only. To explore the effect of allosteric regulation these data need to be compared with membrane embedded substrates (liposomes of different lipid composition containing PI(3,4,5,)P_3_).

---

## [Author Response]

Several points made by the reviewers (see below) should be taken into account in the revised manuscript. The reviewers have found the allosteric mechanism for the effect of the C2 domain based on crystallography and MD are convincing, but it would be helpful if you could provide a movie that portrays this mechanism.

We now provide 5 videos:

Video 1 and Video 2 show the 4 principal motions (eigenvectors) extracted from principal component analysis of the MD simulations of the Ptase and Ptase-C2.

Video 3 and Video 4 show substrate interactions during a part of the simulations with Ptase and Ptase-C2.

Video 5 illustrates the catalytic cycle presented in Figure 9 and the conformational changes between L4-out and L4-in conformations.

In the revision, you should take account of the suggestions by the reviewers, in particular, in the discussion of an effect on k_cat_ and K_M_ as being indicative of allostery (reviewer 1).

We have corrected our statement according to the comment of reviewer 1 (see below, under replies to reviewer #1”).

An important concern that has been raised by the reviewers is that the lipid blot demonstration of lipid affinity is not adequate for the work. These blots are subject to a range of artifacts. It would be better if you could measure the affinity using lipid vesicles and the FRET assay suggested by reviewer 3. This assay has been described by a number of workers (see, for example, PMID 22949682).

As requested, we performed binding experiments with lipid vesicles. We initially tested the FRET assay suggested by reviewer 3. Under replies to reviewer #3 we explain and show data indicating that this assay is not suitable to measure SHIP interactions with PS vesicles.

We therefore performed lipid vesicle interaction studies by surface plasmon resonance using a Biacore X100 instrument and a L1 sensor chip with immobilized PS vesicles. These data are shown in Figure 2 of the revised manuscript and the lipid blot data is now presented as Figure 2—figure supplement 1.

Also, the methods should be revised to make clear exactly what lipids are being used and how the lipids are prepared for the assay.

We apologize for not specifying the type of lipids used in activity assays. In the previous submission we used soluble short chain lipids PS-diC8 or PC-diC8. This is now clearly stated in the main text (subsection “The C2 domain affects SHIP2 activity”, last paragraph), Figure 4 and its legend and the Materials and methods section. In the revised manuscript we further added experiments with substrate and PS incorporated in vesicles (Figure 4, see point below).

While the reviewers believe that assays with substrate in lipid vesicles would be an informative addition to the manuscript, it may be that this is beyond the scope of the present work, and this should not be regarded as essential.

We agree that this provides an informative addition and have performed activity assays with substrate in vesicles (Figure 4). These data show that the C2 domain also enhances SHIP2 activity on vesicles and that PS incorporated in PIP_3_ vesicles has a larger activating effect than soluble PS-diC8.

In an earlier study of several phosphatases (Nordlund and colleagues), the L4 loop of SHIP2 was referred to as the P4-interacting motif (P4IM). It would be helpful for readers familiar with the previous study if the current manuscript also at least mentions this nomenclature.

In the fifth paragraph of the Discussion we now refer to the P4IM terminology. However, since 2 residues of the P4IM are located outside L4 (for details see replies to reviewer #1”) and because Arg682 in SHIP2 likely has an important role in P3 substrate interactions, we believe it would be misleading to use the P4IM terminology for SHIP throughout the manuscript.

It is claimed at the end of the manuscript that the cellular assays presented show that the C2 domain has an additional role, likely in membrane localization. This statement should be clarified, or if the statement is not based on the results, this should be explained.

This statement was based on the observation that deleting the C2 domain has a larger cellular effect than the FLDD mutations. However, we agree this statement could be misleading. We now simply state: “This confirms that C2 interactions observed in the crystal structure are important for SHIP2 activity in cells.”

The manuscript claims that "both, the Ptase and C2 domains of SHIP2 bind PS in a Ca^2+^ independent manner" This is misleading. The C2 domain on its own was never tested. It could be that the interaction depends entirely on the pptase domain, but that the pptase is only oriented and stabilised by the C2 domain. The statement consistent with the results would be "both the Ptase only and the Ptase-C2 constructs bind PS in a Ca^2+^-independent manner."

We have rewritten the statement as suggested (subsection “The C2 domain of SHIP2”).

Common membrane-binding mechanisms are asp/glu residues binding Ca^2+^, which in turn binds to membranes or lys/arg residues that bind directly to the negatively charged lipids. Figure 2 shows that the SHP2 residues in the key positions are ser or asp/glu residues, but Figure 2 shows that Ca^2+^ does not affect binding. It would be helpful if you briefly discuss this point.

New SPR data have in fact revealed that there is a weak increase of 10% in Ptase-C2 binding to PS in presence of Ca^2+^ (Figure 2). Firstly, we now show in Figure 2 both, acidic and basic residues potentially involved in lipid binding. Further, in the subsection “The C2 domain of SHIP2”, we now discuss that the interaction appears to be dominated by direct binding to basic residues, but the small Ca^2+^ effect we observe could indicate that the weakly conserved acidic residues and serines on CBL1 and 3 of SHIP2 potentially make a small Ca^2+^ dependent contribution to PS binding.

For assistance to color blind readers, it would be helpful if the explanatory keys in Figure 6 are placed next to the curve that they describe, or that there is a line between the curve and the key.

There are shapes for markers on the curves in the figure, but these are typically small in the final version and may not be clear on their own.

From the specific comments of reviewer 2 we understand that the comment concerns Figure 6—figure supplement 1 (now Figure 7—figure supplement 1, according to new numbering). As requested by reviewer 2, we have now clearly marked and labelled the CD curves that show a different behavior.

In Figure 7, since the ratio pAkt/Akt is not the directly observed quantity, the manuscript should make clear how the errors in this ratio were determined. Error propagation in the ratio based on the standard deviations of the two observed quantities should have been carried out. The reviewers are concerned that differences in activity are complicated by differences in expression. To allay these concerns, the statistical methods should be described in the Materials and methods.

We now describe in the Materials and methods section the statistical method used. We do not apply error propagation of the pAkt and total Akt measurements, because the two quantities are not “uncorrelated”, but rather are used for normalisation (see replies to reviewer #2 for more details).

In order to show higher significance that is not related to differences in SHIP expression, we performed 5 additional experiments (each in triplicates), therefore the data we now present in Figure 8 is from 8 independent experiments, each in triplicates (n=24). Taking all data together we still find significance for pAkt/Akt levels, but no significance for differences in SHIP expression levels.

The WT X-ray dataset appears to be the same data set that was described in the earlier crystallisation report, however, the I/σ and highest resolution shell limits are not the same. It appears that the data have been reprocessed, and this is not surprising. However, it should be made clear how the resolution limits were chosen. Since the I/σ for the highest-resolution shells differ for the various datasets, it does not seem that this was the decisive criterion. The correlation coefficients cc1/2 and cc* have been proposed as a criterion to help make resolution cutoff in a statistically reliable manner. The table should also include cc1/2.

We apologize for this and thank the reviewers for pointing this out. Indeed, the data were reprocessed since it was published in the Protein Journal. Different definitions of I/σ in older versions of *scala* (used to cut data) and newer versions of *aimless* (reported in the table) caused the high I/σ values (see replies to reviewer #3 for a more detailed explanation). We have now reprocessed the data to higher resolution and cut the data at a mean (I/sd), reported in *aimless,* of ~2.

CC(1/2) values are now included in Table 1.

Reviewer #1:

[…] In a previous study by Nordlund and colleagues, structures of SHIP2, OCRL, and INPP5B pptase domains were determined and compared. In this previous study, it was shown that a motif referred to as the P4-interacting motif (P4IM) interacts with 3- and 4-phosphates of the substrates. In the manuscript of Lietha and colleagues, the same feature is referred to as loop L4. To simplify things for the readers, it would be helpful if the authors of the current manuscript use the P4IM notation or at least point out that this is contained in L4. The presence of the C2 domain results in a large change in the conformation of the L4 loop, and the authors propose that this change likely constitutes an important component of the catalytic cycle of SHIP2 on membranes.

In Tresaugues et al. (2014) the P4-interacting-motif (P4IM) is defined to be composed of the INPP5B residues Tyr502, Lys503, Arg518, and Lys516. Of these, only Arg518 (Asn684 in SHIP2) and Lys516 (Arg682 in SHIP2) are contained within loop L4. As suggested we point out that residues Asn684 and Arg682 in SHIP2 correspond to residues in the P4IM (Discussion, fifth paragraph). However, since 2 residues of the P4IM are located outside L4 and because Arg682 in SHIP2 likely has an important role in P3 substrate interactions, we believe it would be misleading to use the P4IM terminology in SHIP throughout the manuscript.

In the first paragraph of the subsection “The C2 domain affects SHIP2 activity” it states that changing k_cat_ without much change in K_M_ suggests that the activation is allosteric. The most common definition of allosteric is an effect on activity due to interaction outside the catalytic site. This can be an effect on K_M_ (K-type allostery), or it can be an effect on V_max_ (V-type), or come combination of these. However, simply an effect on k_cat_ does not suggest an influence directed from outside the active site. A k_cat_-only effect could be obtained by forming additional interactions with the transition state, but not with the substrate, but this would not be an allosteric effect. What the authors mean is that if the C2 domain has any influence at all, it must be allosteric, since the C2 domain is not near the active site. This should be restated. There is also a grammatic problem with the sentence in its current form (should be "suggests" instead of "suggesting").

We thank the reviewer for pointing this out. We now simply state “The fact that the distant C2 domain has specific and differential effects on catalysis of the two substrates, suggests the presence of an allosteric communication between the domain interface and the active site.”

In the last paragraph of the subsection “The C2 domain affects SHIP2 activity”, describing Figure 3, it is not clear what the authors have done in the experiments that they describe as adding "PC" and "PS". What exactly are the lipids that they have used? If they are synthetic, they should state what the acyl chains are. If they are from a natural source, they should quote the dominant component of the source. It is particularly important the length and saturation of the acyl chains. Also, there is no description of how the lipid substrates were presented. Were they prepared as unilamellar vesicles? Were the multilammelar vesicles? Were they SUVS? LUVS? GUVS? The reason that this is important is that the authors are using a di-C8 PIP3 lipid that is soluble on its own, so it is not clear that this lipid would effectively and completely partition into membranes of lipid vesicles. If the lipid substrate is both in the bulk phase and in the membranes, it makes interpretation of the results more complex.

We apologize for not specifying the type of lipids used. We now state in the main text of the Results section (subsection “The C2 domain affects SHIP2 activity”, last paragraph), Figure 4 and its legend and the Materials and methods section that initially we used soluble PS-diC8 and PC-diC8 in these experiments. The revised manuscript now also includes data with PS vesicles (new Figure 4, see point below).

Ideally, it would be helpful if the authors had used long-chain PIP3 incorporated into unilamellar vesicles for at least some of their assays, in order to strengthen their conclusions for the behavior of SHIP2 in cells.

As suggested, we performed additional experiments with long-chain PIP3 embedded in vesicles containing 30% PS or PC. The new data are shown in Figure 4 and show that the C2 domain also enhances activity of SHIP in vesicles and this effect is abrogated by FLDD interface mutants. The effect of PS inclusion in vesicles is described in the point below.

In the aforementioned paragraph, it is proposed that PS orients the enzyme on PIP3-containing membranes. This is an interesting proposition that could be tested if the substrate were PIP3 in vesicles, however, the authors are using di-C8 PIP3, which is soluble and not necessarily entirely present in membranes. It is hard to assess what is happening in Figure 3, which is intended to show the effect of PS. In Figure 3, Both the pptase and pptase-C2 constructs are activated by PS, so there is little indication in this that the C2 domain is contributing to the activation by PS. Figure 2 suggests that the pptase-C2 construct binds better to PS than the pptase domain alone. However, from 3G, this does not translate to PS-enhanced activity. This is perhaps not surprising, since the di-C8 PIP3 substrate is soluble and perhaps not fully incorporated into the undescribed PS-containing membranes. This experiment would be far more informative if the study were using PIP3/PC/PS unilamellar vesicles with long-chain fatty acids on all of the lipids. I would not regard these vesicle assays as essential for the paper, but they would enhance the study.

As explained above, we have now performed such studies and show the results in Figure 4. Indeed, when PS is incorporated in vesicles together with the PIP3 substrate, the PS effect is larger than with soluble PS-diC8 and PIP3-diC8. This supports the idea that PS in membranes can orient the SHIP2 enzyme, however, this does not appear to be solely achieved by binding of the C2 domain to PS, since PS also activates the Ptase domain alone. This is consistent with a significant PS binding of the Ptase observed in Figure 2 and Figure 2—figure supplement 1.

In the last paragraph of the subsection “Probing the communication path by mutagenesis”, the authors state that their cellular results confirm that C2 interactions observed in the crystal structure are important for SHIP2 activity in cells and that the C2 domain has an additional role, likely in membrane localization. It is not clear how these results show that the C2 domain has an additional role in cells. This should be clarified or the statement removed.

We agree this statement could be misleading. It was based on cell biology data showing that deletion of C2 has a much stronger effect than the FLDD mutation. However, new data presented in Figure 4 suggest that membrane localization (at least via PS) is not solely due to the C2 domain. Possibly, the intrinsic activating effect of FLDD mutations could contribute to the difference between the DC2 and FLDD mutants in cellular experiments. To avoid any confusion, we now simply state: “This confirms that C2 interactions observed in the crystal structure are important for SHIP2 activity in cells.”

Reviewer #2:

[…] This study includes a very large number of experiments and addresses an important question. However, there are a number of points that could be improved. My suggestions are as follow:

1) Figure 2. I found this figure confusing. The most common mechanisms of binding negatively charged lipids are asp/glu residues binding Ca^2+^ or lys/arg residues that directly to the negatively charged lipids. Figure 2 shows that the SHP2 residues in the key positions are ser or asp/glu residues. There are no positively charged amino acids so one would predict a role for Ca^2+^. But Figure 2 shows that Ca^2+^ does not affect binding. This says to me that Figure 2 is misleading. The binding site must be elsewhere in the C2 domain.

We agree and thank the reviewer for pointing this out. In the revised manuscript we show in Figure 2 both, the acidic residues in CBL1 and 3 and basic residues on C2 and Ptase domains expected to face the membrane. New SPR data have in fact revealed that there is a weak increase in Ptase-C2 binding to PS in presence of Ca^2+^ (new Figure 2), an effect we don’t observe for the Ptase. In the subsection “The C2 domain of SHIP2”, we now discuss that the PS interaction appears to be dominated by direct interactions with basic residues, but possibly the weakly conserved acidic residues and serines on CBL1 and 3 of SHIP2 could be responsible for the small Ca^2+^ dependent contribution to PS binding we observe.

2) Figure 6—figure supplement 1. I am partly color blind. It would be convenient if the figure legend identified the mutants that have different spectra.

We have clearly labelled the curves that show a different behavior. According to the new numbering this is now in Figure 7—figure supplement 1.

3) Figure 7. This experiment examines two mutants in HEK293 cells. This figure surprised me in two ways. The recombinant SHIP2 is a dark band. The endogenous SHIP2 is not detected in the control lane so the SHIP2 is grossly over expressed but there is only a 40% decrease in PAkt.

Actually, we do detect a very weak band corresponding to endogenous SHIP2 in the control lane. However, since SHIP expression with the pOPINJ vector adds an N-terminal GST tag (this is now clearly stated in the Materials and methods section and legend of Figure 8—figure supplement 1), endogenous SHIP2 runs at a lower molecular weight, close to that of the SHIP2 DC2 construct. Nevertheless, it is correct that native SHIP levels appear low and the transfected constructs are strongly overexpressed.

Regarding the limited decrease in pAkt, we believe there could be several reasons explaining this behavior. As described in the Introduction, the effect of SHIP on Akt phosphorylation is complex and likely depends on several other factors, including: 1) Whether the SHIP product PI(3,4)P_2_ is efficiently further degraded to PI(3)P (since PI(3,4)P_2_ has still partial activity towards Akt); 2) What the relative contributions are of SHIP and PTEN in PIP3 degradation under the chosen condition and cellular context and how these activities compare to PI3K activity generating PIP3; 3) How efficiently is the overexpressed SHIP2 targeted to the membrane under the experimental conditions? We believe any of these factors could cause a limited effect of the expressed SHIP proteins on reducing Akt phosphorylation.

Second, the deletion of the C2 domain clearly reduces activity, but the FLDD mutation has only a minute affect, which they say is statistically significant. They do not explain the statistical analysis. How was the error for PAkt propagated with the error in Akt?

We do not apply error propagation of the pAkt and total Akt measurements, because the two quantities are not “uncorrelated”. The typical error propagation for a ratio Q according to the formula:

(in our case Q=pAkt/Akt; δ(X)=error in measurement of X, pAkt or Akt)

assumes that the quantities are uncorrelated. However, in our case dividing by total Akt is a “normalization” of the phospho-Akt signal and only the ratio is expected to have similar values, not necessarily the individual values. To our knowledge, and after talking to several cell biologists in our institute, it is standard practice in cell biology to calculate errors on the ratio of normalized quantities, without error propagation.

I am just not convinced that the small effect of the FLDD mutation is real and not due to error or differences in expression. They have a supplemental figure for expression. The FLDD is slightly lower but they say that it is not statistically significant. I need a little bit more convincing, and the statistics should be in Materials and methods.

In order to show higher significance that is not related with differences in SHIP expression, we performed an additional 5 independent experiments, each in triplicates (i.e. total 8 experiments, n=24). Admittedly, the only partial reduction in pAkt levels upon SHIP expression limits the window showing clear significance and the FLDD mutations having only a partial reduction in activity further limits the significance. Nevertheless, after 8 triplicate experiments (n=24), our results do still show significance between WT and FLDD mutants. Taking all experiments together the total SHIP expression levels are no longer smaller for the FLDD mutant (actually slightly higher, but without significance), making it unlikely that differences in pAkt levels are due to different SHIP expression levels. We would also like to point out that in the revised manuscript we include activity data with PIP3 embedded in vesicles (Figure 4) that further support that FLDD mutations affect SHIP activity in a native membrane environment.

We now clearly state in the Materials and methods section how we analyze the statistics for cell biology experiments, i.e. that standard errors are calculated on values of pAkt/Akt ratios and significance is calculated using a two-tailed unpaired Student t test assuming Gaussian distribution.

Reviewer #3:

Le Coq et al. Report the crystal structure of a SHIP2 construct that, unlike previously reported structures, contains the membrane targeting C2 domain. The reported structural analysis together with molecular dynamics simulations reveals how the C2 domain (and thus membrane binding) allosterically controls the enzymatic activity of SHIP2. The findings are important, most of experiments are well done, manuscript is well written and the reasoning is sound and square. The manuscript clearly deserves to be published in eLife. However, there is one major issue (and several small technical issues) that should be improved before final acceptance. The additional control experiments should not take more than a day or two of a skilled experimentalist.

1) A protein lipid overlay assay is a useful screening technique, however, no conclusions should be derived from this method as it is notoriously known to be artifact prone. The authors need to verify their results using liposomes with/out PS and with/out Ca^2+^. Many reliable standard assays exist. Since SHIP2 contains trypthophan residues I suggest to use a FRET based assay (W residues as donors and dansyl-PE as acceptors) to measure binding.

As requested, we performed binding experiments with lipid vesicles. We initially tested the suggested FRET assay using energy transfer between excited tryptophans and dansyl-PE. However, instead of a FRET induced increase at the dansyl emission in presence of protein, we observed a decrease (see Figure 10). The reason for is not exactly clear to us, but appears to result from quenching of liposome fluorescence by protein (apparently also observed at low level in Baskaran et al. (2012) Mol Cell 47, 339-48). As a result, this assay is not suitable to measure SHIP interactions with PS vesicles.

Author response image 1.Tryptophan-dansyl FRET assay.(**A**) Emission scan resulting from excitation at 280nm of dansyl vesicles alone (blue trace) or vesicles in presence of SHIP2 Ptase (red trace). SHIP2 Ptase induces a reduction of the emission signal at the dansyl emission (maximum at 513nm). (**B**) Emission of dansyl vesicles alone (I_0_ set to 100%; blue bars) or in presence of SHIP2 Ptase or PtaseC2 (as% of I_0_; red bars). (**C**) The emission at 513nm of dansyl vesicles decreases linearly with increasing concentrations of SHIP2 Ptase.**DOI:**
http://dx.doi.org/10.7554/eLife.26640.036

In an alternative approach, we performed lipid vesicle interaction studies by surface plasmon resonance (SPR) using a Biacore X100 instrument and a L1 sensor chip with immobilized PS and PC vesicles. These data are shown in Figure 2 of the revised manuscript and the lipid blot data is now presented as Figure 2—figure supplement 1. Shown are SPR responses resulting from differences between the active flow cell coated with 30% (mol/mol) PS vesicles and the reference flow cell containing 100% PC vesicles. These data confirm a stronger interaction with Ptase-C2 comparted to the isolated Ptase, although the difference is smaller compared to PLO experiments. Unfortunately, we were not able to extract dissociation constants or kinetic k_on_/k_off_ values from SPR sensorgrams. Binding and dissociation appear to follow non-trivial kinetics, which we were not able to fit with standard models. On the other hand, plotting steady-state values versus protein concentration suggests that K_D_ values are significantly above the highest protein concentration used (10µM).

Injecting higher protein concentrations caused irreversible deposition of protein on the surface making responses unreliable. Interestingly, using injections at 5 µM protein we detected a weak but significant effect of Ca^2+^ only with Ptase-C2, increasing the binding by ~10%. We do not claim that this modest effect is biologically significant, but as discussed in the subsection “The C2 domain of SHIP2”, this effect might result from the low conservation of Ca^2+^ binding residues in CBL1 and 3 of the SHIP2 C2 domain.

2) Biochemical (enzymatic) analysis is done using soluble substrates only. To explore the effect of allosteric regulation these data need to be compared with membrane embedded substrates (liposomes of different lipid composition containing PI(3,4,5,)P_3_).

As suggested, we performed enzyme activity assays with vesicles containing PIP3. These data are presented in Figure 4 of the revised manuscript. The data show that the C2 domain also promotes SHIP2 activity on vesicles and this effect is removed by the FLDD interface mutants. We further show that PS in vesicles promotes SHIP activity in an independent and additive manner to the C2 effect.